# Asymmetric division triggers cell-specific gene expression through coupled capture and stabilization of a phosphatase

**Niels Bradshaw, Richard Losick***

Department of Molecular and Cellular Biology, Harvard University, Cambridge, United States

**Abstract** Formation of a division septum near a randomly chosen pole during sporulation in *Bacillus subtilis* creates unequal sized daughter cells with dissimilar programs of gene expression. An unanswered question is how polar septation activates a transcription factor ($\sigma^F$) selectively in the small cell. We present evidence that the upstream regulator of $\sigma^F$, the phosphatase SpoIIE, is compartmentalized in the small cell by transfer from the polar septum to the adjacent cell pole where SpoIIE is protected from proteolysis and activated. Polar recognition, protection from proteolysis, and stimulation of phosphatase activity are linked to oligomerization of SpoIIE. This mechanism for initiating cell-specific gene expression is independent of additional sporulation proteins; vegetative cells engineered to divide near a pole sequester SpoIIE and activate $\sigma^F$ in small cells. Thus, a simple model explains how SpoIIE responds to a stochastically-generated cue to activate $\sigma^F$ at the right time and in the right place.

*For correspondence: losick@mcb.harvard.edu

## Introduction

How genetically identical daughter cells adopt dissimilar programs of gene expression following cell division is a fundamental problem in developmental biology. A common mechanism for establishing cell-specific gene expression is asymmetric segregation of a cell fate determinant between the daughter cells (*Horvitz and Herskowitz, 1992*; *Neumüller and Knoblich, 2009*). In polarized cells, intrinsic asymmetry can be inherited from generation to generation. For example, the dimorphic bacterium *Caulobacter crescentus* localizes certain cell fate determinants to the old cell pole, leading to their asymmetric distribution following division (*Iniesta and Shapiro, 2008*; *Bowman et al., 2011*). However, non-polarized cells such as *Bacillus subtilis* must generate asymmetry de novo, which is passed on to the daughter cells to differentiate.

*Bacillus subtilis* divides by binary fission to produce identical daughter cells during vegetative growth but switches to asymmetric division when undergoing the developmental process of spore formation (*Piggot and Coote, 1976*; *Stragier and Losick, 1996*). To sporulate, cells place a division septum near a randomly chosen pole of the cell (*Veening et al., 2008*) to create two unequally sized daughter cells with dissimilar programs of gene expression. The smaller cell, the forespore, which largely consists of the cell pole, will become the spore, whereas the larger cell, the mother cell, nurtures the developing spore (*Figure 1B*). An enduring mystery of this developmental system is how stochastically generated asymmetry initiates dissimilar programs of gene expression in the daughter cells resulting from polar division (*Barak and Wilkinson, 2005*).

The earliest acting cell-specific regulatory protein in the sporulation program is the transcription factor $\sigma^F$. The $\sigma^F$ factor and the proteins that control it – SpoIIAB, SpoIIAA, and SpoIIE – are produced at the onset of sporulation (*Gholamhoseinian and Piggot, 1989*), but $\sigma^F$ is held inactive until the completion of asymmetric cell division, when it turns on gene expression selectively in the

**eLife digest** An important question in biology is how genetically identical cells activate different sets of genes. This is particularly perplexing for cells that rely on random events to specify the genes they switch on. Normally, cells of a bacterium called *Bacillus subtilis* divide symmetrically to produce two identical cells that express identical sets of genes. However, *B. subtilis* cells can also undergo a developmental program to form a spore to help it survive periods of extreme conditions. To do this, first a *B. subtilis* cell divides asymmetrically by placing the site of division close to a randomly selected end of the cell. This creates a smaller cell that becomes the spore and a larger cell that nurtures the developing spore. Each cell must turn on different genes to play its role in spore development, but how asymmetry in the position of cell division leads to these differences in gene expression has been a longstanding mystery.

Bradshaw and Losick studied a regulatory protein called SpoIIE, which is responsible for switching on genes in the small cell. SpoIIE is made before cells divide asymmetrically, but only accumulates in the small cell. The experiments revealed that an enzyme broke down the SpoIIE protein if it wasn't in the small cell. This prevented SpoIIE from incorrectly switching on genes before division was completed or in the large cell.

Protection of SpoIIE from being broken down in the small cells was then shown to be linked to the placement of cell division; SpoIIE first accumulates at the asymmetrically positioned cell division machinery and then is transferred to a secondary binding site at the nearby end of the cell. Capture of SpoIIE at the end of the cell was coupled to its stabilization as SpoIIE molecules interacted with one another to form large complexes.

Together these findings provide a simple mechanism to link the asymmetric position of cell division to differences in gene expression. Future studies will focus on understanding how SpoIIE is captured at the end of the cell and how this prevents SpoIIE from being degraded.

forespore (*Margolis et al., 1991*; *Stragier and Losick, 1996*) (*Figure 1A,B*). SpoIIAB is an anti-sigma factor that traps $\sigma^F$ in an inactive complex (*Min et al., 1993*; *Duncan and Losick, 1993*). Escape from SpoIIAB is mediated by the anti-anti-sigma factor SpoIIAA (*Diederich et al., 1994*). SpoIIAA is, in turn, activated by SpoIIE, a member of the PP2C family of protein phosphatases (*Bork et al., 1996*; *Levdikov et al., 2011*). SpoIIE converts the inactive phosphorylated form of SpoIIAA (SpoIIAA-P) to the active dephosphorylated form (*Duncan et al., 1995*) (*Figure 1A*). Dephosphorylation of SpoIIAA-P by SpoIIE is therefore the critical event in activating $\sigma^F$. Understanding how SpoIIE reads out cellular cues to delay dephosphorylation of SpoIIAA-P until after septation and restrict phosphatase activity to the forespore is thus the central challenge in understanding how cell-specific gene transcription is established during sporulation.

SpoIIE consists of three domains: a PP2C phosphatase domain at the C-terminus, a ten-pass transmembrane domain at the N-terminus, and a 270-amino acid central domain (henceforth referred to as the regulatory domain) that is important for regulating SpoIIE compartmentalization and activity (*Figure 2B; Arigoni et al., 1999*). Prior to asymmetric cell division, SpoIIE localizes to the polar divisome and contributes to its placement (*Arigoni et al., 1995*; *Ben-Yehuda and Losick, 2002*). After septation is complete, SpoIIE is found principally in the forespore and to a limited extent at a second polar divisome near the distal cell pole (*Figure 1C*, *Video 1*).

Here we describe three interdependent features of SpoIIE that together explain how SpoIIE links polar septation to the cell-specific activation of $\sigma^F$: (1) SpoIIE is proteolytically unstable and is degraded dependent on the AAA+ protease FtsH; (2) SpoIIE is transferred from the polar divisome – the de novo origin of asymmetry – to the proximal cell pole during polar septation; and (3) SpoIIE forms homooligomeric complexes which promote capture at the pole, protection from proteolysis and activation as a phosphatase, thus linking the cues that direct localization of SpoIIE to its stabilization and activation.

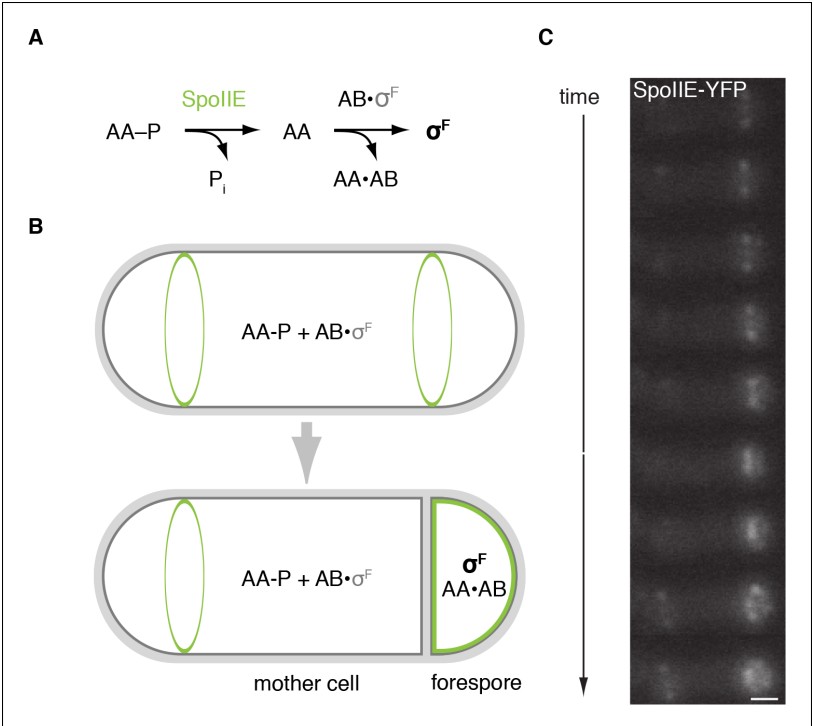

**Figure 1.** SpoIIE is compartmentalized in the forespore and activates σ$^F$. (**A**) Diagram of the pathway for σ$^F$ activation. Phorphorylated SpoIIAA (AA-P) is dephosphorylated by SpoIIE. Dephosphorylated SpoIIAA (AA) then binds to SpoIIAB (AB) displacing σ$^F$ and leading to σ$^F$-directed transcription. (**B**) SpoIIE (green), AA, AB and σ$^F$ are produced in predivisional cells. Prior to completion of asymmetric cell division SpoIIE associates with the polar divisome near one or both cell poles (the pole at which division initiates is chosen randomly). Following completion of cytokinesis, SpoIIE is enriched in the forespore where it dephosphorylates AA-P to activate σ$^F$. (**C**) A montage of images taken every 6 min from a single sporulating cell (strain RL5876) producing SpoIIE-YFP. Cells are oriented with the forespore on the right as in the diagram. A movie of this sporulating cell is provided as *Video 1*. Scale bar: 0.5 μm.

The following figure supplements are available for Figure 1:

**Figure supplement 1.** SpoIIE constricts along with FtsZ during asymmetric cell division.

# Results

## SpoIIE is degraded in an FtsH-dependent manner

Although transcription of s*poIIE* commences in pre-divisional cells and continues in the mother cell following cytokinesis (*Fujita and Losick, 2003*), SpoIIE protein and activity are restricted to the forespore (*Figure 2A*). This apparent contradiction led us to consider the possibility that spatially restricted proteolysis contributes to compartmentalization of SpoIIE. Selective stabilization of SpoIIE in the forespore coupled to efficient global degradation would enrich SpoIIE in the forespore despite ongoing transcription of *spoIIE* in predivisional cells and the mother cell. To investigate this hypothesis, we sought to determine if SpoIIE turns over on a timescale commensurate with σ$^F$ activation and, if so, to identify the responsible protease.

To detect SpoIIE degradation during sporulation, we monitored the disappearance of SpoIIE

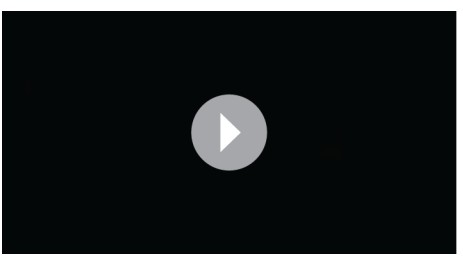

**Video 1.** Movie file of the sporulating cell shown in *Figure 1C* (2fps).

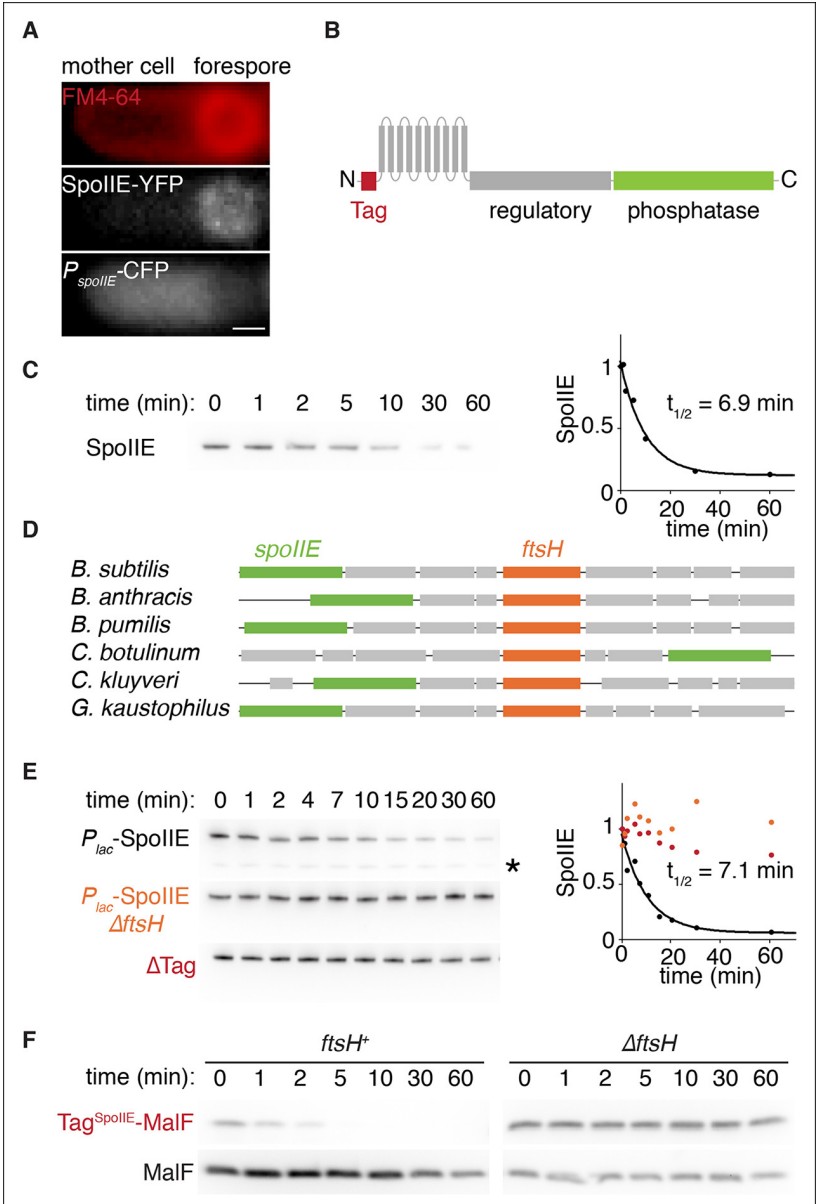

**Figure 2.** SpoIIE degradation depends on FtsH. (**A**) SpoIIE is compartmentalized to the forespore. A single sporulating cell (strain RL5874) is shown following the completion of asymmetric septation. The membrane stained with FM4-64 is shown in red above SpoIIE-YFP and CFP driven by an in frame fusion to the start of the *spoIIE* open reading frame. Scale bar: 0.5 μm. (**B**) The domain architecture of SpoIIE. The N-terminal cytoplasmic tail (red), followed by 10 transmembrane-spanning segments, the regulatory region (amino acids 320–589, gray), and the phosphatase domain (amino acids 590–827, green). (**C**) SpoIIE is degraded during sporulation. Translation was arrested (by addition of 100 μg/ml chloramphenicol) in sporulating cells producing SpoIIE-FLAG (strain RL5877), and samples were withdrawn at the indicated times. SpoIIE was detected by western blot using α-FLAG monoclonal antibody (left). Quantitation of the western (right) fit to a single exponential equation. (**D**) The genes for s*poIIE* and *ftsH* are near each other in the genome with conserved synteny. The diagram shows genomic organization of diverse endospore forming species. Filled boxes indicate genes with s*poIIE* in green, *ftsH* in orange, and other genes in gray. (**E**) SpoIIE degradation requires FtsH, and FtsH mediated degradation requires the N-terminal cytoplasmic tail (Tag^SpoIIE) of SpoIIE. Translation was arrested in vegetatively growing cells producing SpoIIE-FLAG with an IPTG inducible promoter (strain RL5878 wt, RL5879 *ΔftsH*, and RL5880 SpoIIE-ΔTag (residues 11–37 were deleted). Degradation was monitored (left) and quantitated (right) as in panel C. (**F**) Tag^SpoIIE is sufficient to target a heterologous protein for FtsH-dependent degradation. Degradation of MalF-TM-FLAG fused at the N-terminus to either Tag^SpoIIE (wt RL5888, *ΔftsH* RL5889) or the first 10 amino acids of SpoIIE (wt RL5890, *ΔftsH* RL5891) produced during exponential growth was monitored as in panel C.

The following figure supplements are available for Figure 2:

**Figure supplement 1.** SpoIIE degradation depends on FtsH.

with a C-terminal FLAG tag following inhibition of translation with chloramphenicol (*Figure 2C*). SpoIIE-FLAG was degraded with a half-life of 7 min, demonstrating that SpoIIE is unstable relative to the approximately 1 hr progression of asymmetric cell division and $\sigma^F$ activation (*Figure 1C*) and supporting spatially restricted degradation as a plausible mechanism to compartmentalize SpoIIE.

Next, we sought to identify the protease that degrades SpoIIE. We noticed that the gene (*ftsH*) for the transmembrane AAA+ protease FtsH is located near *spoIIE* in the genome with highly conserved synteny (*Figure 2D*). Furthermore, FtsH is known to degrade transmembrane protein substrates (*Akiyama, 2009*), making it an attractive candidate protease for SpoIIE. FtsH degrades several proteins that block entry into sporulation and prevent the expression of *spoIIE*, such as the Spo0A inhibitor Spo0E (*Le and Schumann, 2009*). We therefore engineered the synthesis of SpoIIE during vegetative growth to bypass the requirement for FtsH in the expression of *spoIIE*. In exponential phase cells deleted for *ftsH*, SpoIIE was stable for more than 1 hr after chloramphenicol treatment, whereas in *ftsH* cells SpoIIE was degraded as rapidly as during sporulation ($t_{1/2}$ = 7.1 min) (*Figure 2E*). (SpoIIE instability and its dependence on FtsH was also seen with untagged SpoIIE [*Figure 2—figure supplement 1A,B*]). We conclude that SpoIIE is degraded in an FtsH-dependent manner. The simplest explanation for this is that SpoIIE is a direct substrate for the protease.

Finally, we attempted to identify the feature or features of SpoIIE that renders it susceptible to proteolysis by FtsH. Truncation of the N-terminal, cytosolic tail of SpoIIE (removal of residues 11 to 37) blocked degradation (*Figure 2E*, ΔTag), whereas removal of the regulatory domain or the phosphatase domain or substitution of the transmembrane domain with the first two transmembrane segments of *E. coli* MalF (MalF-TM) did not impede FtsH-dependent degradation (*Figure 2—figure supplement 1C*). Additionally, the first 37 amino acids of SpoIIE (Tag$^{SpoIIE}$) were sufficient to confer FtsH-dependent degradation on a heterologous protein, MalF-TM-FLAG (*Figure 2F*). Therefore, the N-terminal tail of SpoIIE is a tag that is both necessary and sufficient for FtsH-dependent proteolysis.

## Degradation restricts SpoIIE and $\sigma^F$ activity to the forespore

To test whether SpoIIE degradation is required for compartmentalization of $\sigma^F$ activity and SpoIIE, we examined the effect of blocking degradation during sporulation. Here and in the experiments that follow, we removed Tag$^{SpoIIE}$ from SpoIIE in cells that were wild type for *ftsH* to selectively block SpoIIE degradation and circumvent off-target effects from other FtsH substrates had we used an *ftsH* mutation. Indeed, we observed a dramatic increase in aberrant activation of $\sigma^F$ in *Δtag spoIIE* cells (*Figure 3A*). Whereas in wild-type cells $\sigma^F$ activity was highly specific for the forespore (less than 2% non-specific activation), *Δtag spoIIE* caused non-specific activation of $\sigma^F$ in 71% of the cells (*Figure 3B*). Quantification of $\sigma^F$ activity with a *lacZ* reporter revealed that a strain with *Δtag spoIIE* activated $\sigma^F$ with a similar time dependence but had 10-fold elevated $\sigma^F$ activity (*Figure 3C*).

Activation of $\sigma^F$ is tightly coupled to the completion of asymmetric cell division, and SpoIIE mutants have been characterized that uncouple cell division and $\sigma^F$ activation (*Carniol et al., 2004*; *Feucht, et al., 2002*; *Hilbert and Piggot, 2003*). In contrast to these other cases of $\sigma^F$ mis-activation in predivisional cells, stabilization of SpoIIE led to activation of $\sigma^F$ primarily in cells that had completed asymmetric cell division (*Figure 3A,D*). Thus, stabilization of SpoIIE only partially uncouples $\sigma^F$ activation from cell division.

Consistent with the idea that degradation contributes to compartmentalization of $\sigma^F$ activity by helping to restrict SpoIIE to the forespore, we observed a striking correlation between elevated levels of ΔTag-SpoIIE in the mother cell and mis-activation of $\sigma^F$ (*Figure 3A* bottom panels). Together, these data indicate that SpoIIE degradation is required for compartmentalization both of SpoIIE and $\sigma^F$ activity.

## SpoIIE mutants blocked in compartmentalization and $\sigma^F$ activation

How is SpoIIE selectively stabilized in the forespore? We considered two models: (1) FtsH is not active in the forespore, or (2) specific features of SpoIIE stabilize it in the forespore. To test the former possibility, we engineered the production of the model FtsH substrate Tag$^{SpoIIE}$-MalF using a forespore specific, $\sigma^F$-dependent promoter. Tag$^{SpoIIE}$-MalF was rapidly degraded in a manner dependent on the Tag$^{SpoIIE}$ (*Figure 3—figure supplement 1*). Therefore FtsH is active in the forespore, suggesting that SpoIIE is specifically stabilized against FtsH-dependent degradation.

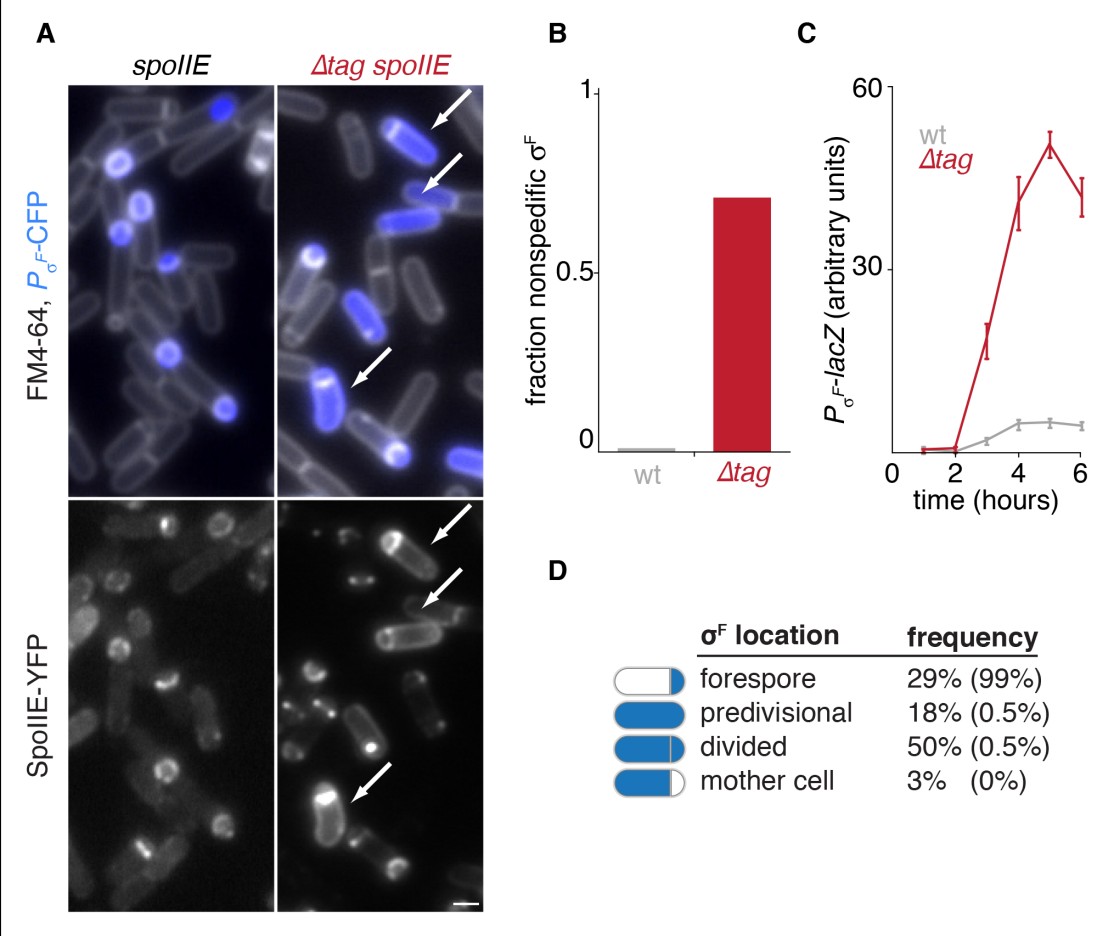

**Figure 3.** Degradation of SpoIIE is required to compartmentalize SpoIIE and σ$^F$ activity. (**A**) Images of sporulating cells producing SpoIIE-YFP (left, strain RL5876) or ΔTag SpoIIE-YFP (right, strain RL5892). Top images show CFP (blue) produced under the control of the σ$^F$ dependent *spoIIQ* promoter and FM4-64-stained membrane (white); bottom images show SpoIIE-YFP. White arrows indicate cells with uncompartmentalized σ$^F$ activity and SpoIIE in the mother cell. The contrast for images of SpoIIE-YFP has been adjusted to approximately 5X brighter than for ΔTag-SpoIIE-YFP for display purposes. Scale bar: 1 μm. (**B**) Quantification of the forespore specificity of σ$^F$ activity from hundreds of cells from images as shown in panel A. (**C**) σ$^F$ activity was measured during sporulation using a translational fusion of the σ$^F$ dependent SpoIIQ promoter to LacZ (wt SpoIIE strain RL5893, ΔTag SpoIIE strain RL5894). Time after initiation of sporulation is indicated, and error bars represent the standard deviation from three biological replicates. (**D**) Quantification of the dependence of σ$^F$ activation on asymmetric cell division driven by ΔTag-SpoIIE. Hundreds of cells from images as shown in panel A were manually assessed for completion of asymmetric cell division and compartmentalization of σ$^F$ activity. The percent of cells with each pattern of σ$^F$ activity is indicated for *Δtag-spoIIE* cells (values for wt *spoIIE* cells are indicated in parenthesis).

The following figure supplements are available for Figure 3:

**Figure supplement 1.** FtsH is active in the forespore.

To identify features of SpoIIE required for its accumulation in the forespore, we screened for SpoIIE variants defective in compartmentalization. We created amino acid substitutions of the most highly conserved residues in SpoIIE and tested these variants (and previously described variants) for function in sporulation (*Figure 4A*, *Figure 4—source data 1*) and forespore accumulation (*Figure 4B*). To monitor accumulation in the forespore of each SpoIIE variant, we compiled average profiles of SpoIIE-YFP along the long axis of hundreds of cells that had undergone polar division. Through this analysis, we identified nine variants of SpoIIE (for example SpoIIE$^{K356D}$) that were absent in the forespore (*Figure 4A,B* blue) and accumulated to reduced levels (*Figure 4C*). Variants with normal compartmentalization, in contrast, accumulated at approximately wild-type levels (*Figure 4B,C* black). Supporting the idea that failure to accumulate in the forespore was due to unrestricted, FtsH-dependent degradation, removal of Tag$^{SpoIIE}$ restored these SpoIIE mutant proteins to

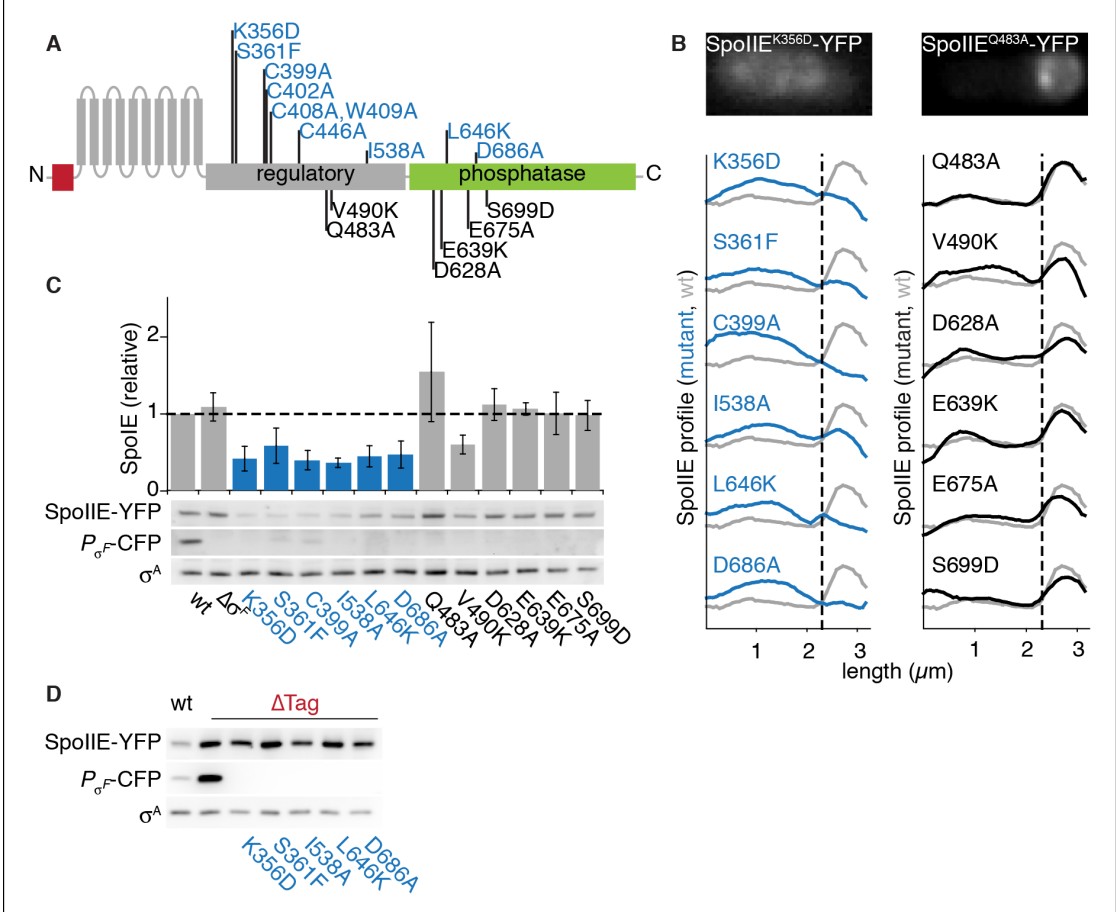

**Figure 4.** SpoIIE stabilization and localization mutants. (**A**) Diagram of SpoIIE mutants with sporulation defects. Variants with localization and $\sigma^F$ activation defects are shown above the diagram in blue, and variants with normal localization but defects in $\sigma^F$ activation are shown below the diagram in black. (**B**) Localization of SpoIIE mutants. Hundreds of asymmetrically divided cells were aligned at the forespore pole to generate average profiles of SpoIIE-YFP localization for each SpoIIE mutant (strains RL5895- 5909) with a reference plot (gray) from wild-type SpoIIE-YFP from $\sigma^F$ mutant cells (strain RL5910). The dashed line represents the approximate position of the asymmetric septum. Images of representative cells with the mislocalized variant SpoIIE$^{K356D}$ (left, strain RL5895) and forespore-localized SpoIIE$^{Q483A}$ (right, strain RL5904) are shown. (**C**) Western blots of protein levels in SpoIIE mutant strains shown in panel B probed for SpoIIE-YFP (with $\alpha$-GFP antibody), CFP produced under the control of a $\sigma^F$-driven promoter, and $\sigma^A$ as a loading control. Levels of each SpoIIE variant were normalized to wt SpoIIE (strain RL5876). Error bars represent the standard deviation from three biological replicates. All localization mutants (shown in blue) are different from the $\sigma^F$ mutant control with p values less than 0.0025 from a paired t-test. (**D**) Western blots of SpoIIE compartmentalization mutants with Tag$^{SpoIIE}$ removed (strains RL5911-5916, with strain RL5876 as a reference) as in panel C.

The following source data and figure supplements are available for figure 4:

**Source data 1.** Sporulation efficiency of SpoIIE mutants.

**Figure supplement 1.** An allele specific suppressor of SpoIIE$^{K356D}$ rescues compartmentalization and stabilization.

---

levels several-fold higher than for wild-type SpoIIE and equivalent to ΔTag SpoIIE (**Figure 4D**). We conclude that SpoIIE undergoes a transition in the forespore that protects it from FtsH-dependent proteolysis and that this transition is blocked by amino acid substitutions such as K356D.

Additionally, we tested whether stabilization of the SpoIIE mutant proteins was sufficient to support $\sigma^F$ activation independent of their susceptibility to degradation. We found that even when Tag$^{SpoIIE}$ was removed, the mutant proteins failed to activate $\sigma^F$ (**Figure 4D** middle panel). A simple unifying model is that the proposed, K356-dependent conformational rearrangement that protects SpoIIE from proteolysis in the forespore is also required to allow it to activate $\sigma^F$.

## An allele-specific suppressor of SpoIIE$^{K356D}$

To investigate the link between stabilization, compartmentalization and activation of SpoIIE, we selected for and isolated several suppressors that restored sporulation to the compartmentalization-defective mutant *spoIIE$^{K356D}$*. We chose this mutant because the K356D substitution was located in the regulatory domain of SpoIIE and caused a particularly severe sporulation defect. We isolated intragenic suppressors at two codons in an apparently saturating screen (see Materials and methods, *Figure 4—source data 1* and *Figure 4—figure supplement 1*). One of them, causing a T353I substitution, was allele-specific (it suppressed *spoIIE$^{K356D}$* but not *spoIIE$^{S361F}$* or *spoIIE$^{V490K}$*, *Figure 4— source data 1*). The T353I substitution restored SpoIIE$^{K356D}$ to wild-type protein levels (*Figure 4— figure supplement 1B*), partially restored restriction of SpoIIE$^{K356D}$ to the forespore (*Figure 4—figure supplement 1C*), and restored compartment-specific σ$^F$ activation (*Figure 4—figure supplement 1D*). The coordinated rescue of these phenotypes by a single amino acid substitution supports our model that a common feature of SpoIIE mediates protection from proteolysis, accumulation in the forespore, and activation of σ$^F$.

The other intragenic suppressors of SpoIIE$^{K356D}$ were substitutions at V697 (V697A and V697F), which is located in the phosphatase domain of SpoIIE. V697A had been independently isolated previously and shown to cause premature activation of σ$^F$ (in the absence of the K356D substitution) (*Hilbert and Piggot, 2003*). These suppressors were not allele specific; V697A suppressed all other mutants of SpoIIE, including the compartmentalization defective SpoIIE$^{S361F}$ mutant and the compartmentalized SpoIIE$^{Q483A}$ mutant (*Figure 4—source data 1*, [*Carniol et al., 2004*]). The V697A substitution did not restore compartmentalization or stabilization of SpoIIE. It did restore σ$^F$ activation but not compartmentalization of σ$^F$ activity (*Figure 4—source data 1* and *Figure 4—figure supplement 1*). All together, these results suggest that the V697A substitution locks the phosphatase domain in a high activity state, bypassing the activation defect of SpoIIE$^{K356D}$. Indeed, biochemical experiments showed that the V697A substitution enhanced the activity of SpoIIE in dephosphorylating SpoIIAA-P (*Figure 4—figure supplement 1E*).

## SpoIIE is compartmentalized and stabilized by binding to the cell pole

To further investigate the mechanism of SpoIIE compartmentalization, we took advantage of the compartmentalization-defective variant SpoIIE$^{K353D}$ and revisited the localization of stabilized SpoIIE. To isolate events prior to σ$^F$ activation, and because certain targets of σ$^F$ (e.g. *spoIIQ*) affect the localization of SpoIIE (*Campo et al., 2008*), we used a mutant lacking σ$^F$ to analyze the localization of SpoIIE, ΔTag-SpoIIE, and its K353D mutant derivative (in contrast to the experiment of *Figure 3A* in which cells were σ$^{F+}$).

Our most striking observation was that ΔTag-SpoIIE was noticeably enriched at the poles of cells that had not initiated polar division (*Figure 5A*). Polar enrichment was dependent on stabilization by removal of Tag$^{SpoIIE}$ (*Figure 5A* gray line). Because the forespore is derived from the cell pole, we hypothesized that the pole is a landmark that directs SpoIIE compartmentalization. In support of this idea, polar localization was abolished by the K356D substitution and partially restored by the T353I suppressor (*Figure 5A* lower panel). Thus, the pole is a cue that directs compartmentalization of SpoIIE, and the same feature(s) of SpoIIE that is required for polar recognition is also required for stabilization and σ$^F$ activation.

We next asked whether SpoIIE has an intrinsic affinity for the cell pole or whether SpoIIE is captured there by features unique to cells undergoing sporulation. To address this question, we engineered the synthesis of SpoIIE-YFP and ΔTag SpoIIE-YFP in vegetative cells that were blocked in divisome formation through the use of the FtsZ inhibitor MciZ (*Handler et al., 2008*). We observed that ΔTag SpoIIE-YFP was enriched at the ends of these cells, and this localization recapitulated the features of polar localization seen during sporulation: polar localization was only observed for stabilized SpoIIE, was blocked by K356D substitution, and was restored by the T353I suppressor (*Figure 5B*). Therefore, a fundamental, constitutive feature of the cell pole mediates SpoIIE polar localization.

We next sought to identify the feature of the pole that is responsible for SpoIIE localization. DivIVA recognizes the negative curvature of the cell pole and directs the polar localization of several other proteins during growth and sporulation (*Lenarcic et al., 2009*; *Ramamurthi and Losick, 2009*). Recently, DivIVA was shown to co-immunoprecipitate with SpoIIE, making it an attractive

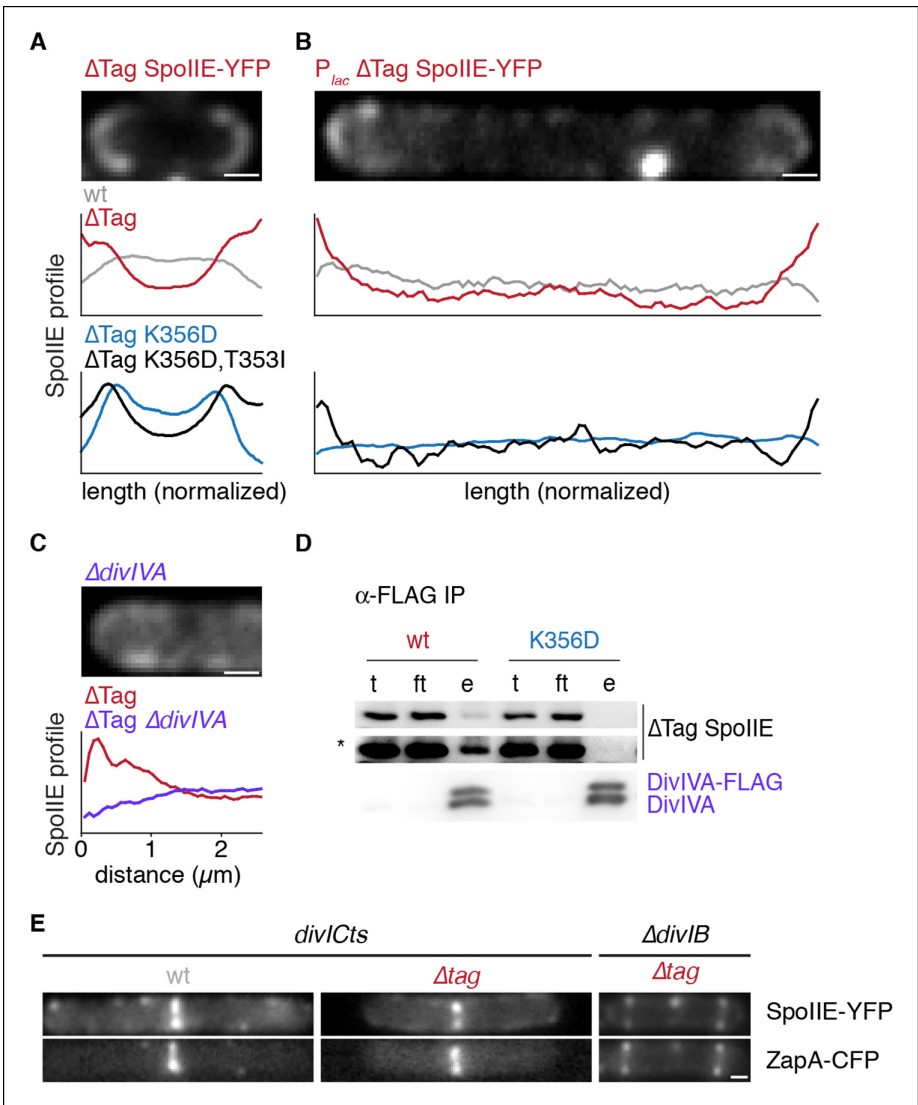

**Figure 5.** Stabilized SpoIIE localizes to the cell pole. (**A**) Average profiles of SpoIIE-YFP from undivided sporulating cells lacking σ^F activity are shown (strains RL5910, 5917, 5912, 5918), with a representative cell with ΔTag-SpoIIE-YFP (strain RL5917) displayed above. (**B**) Vegetatively growing cells expressing SpoIIE-YFP (strains RL5919-5922) displayed as in panel A; variants as indicated) were imaged 30 min after induction of expression of the FtsZ polymerization inhibitor MciZ. (**C**) MciZ-expressing cells (Δ*divIVA* [strain RL5923] or otherwise wildtype [strain RL5924]) were imaged as in panel B and average profiles of ΔTag-SpoIIE-YFP were generated from 20 randomly selected cell poles. (**D**) DivIVA-FLAG was immunoprecipitated with α-FLAG magnetic beads from extracts of sporulating cells expressing SpoIIE variants as indicated (strains RL5925, 5926), and detected by Western blot. The elution (e) shown is 100X concentrated relative to the load (l) and flowthrough (ft) samples. Blots were probed with α-GFP antibody (top two images; lower image in high contrast*) and α-DivIVA antisera (below). Because DivIVA oligomerizes, untagged DivIVA is also co-immunoprecipitated. (**E**) SpoIIE preferentially localizes to the divisome rather than the pole. Representative cells expressing SpoIIE-YFP and CFP-ZapA are shown. Left images show exponentially growing *divICts* cells (strains RL5927, 5928), and right images show a sporulating Δ*divIB* cell (strain RL5929). Scale bars indicate 0.5 μm in all panels.

The following figure supplements are available for Figure 5:

**Figure supplement 1.** Transcription of *spoIIE* in the forespore is not required to compartmentalize SpoIIE.

candidate to anchor SpoIIE to the cell pole (*Eswaramoorthy et al., 2014*). We found that whereas wild-type SpoIIE co-immunoprecipitated with DivIVA from extracts of sporulating cells (as previously observed), SpoIIE^K356D did not (*Figure 5D*). Additionally ΔTag SpoIIE-YFP polar localization during vegetative growth was abolished by a *divIVA* deletion (in the background of a *minD* deletion to suppress the cell division defect of *divIVA* deletion) (*Figure 5C*). Thus, DivIVA directly or indirectly anchors SpoIIE at the cell pole and can do so independently of sporulation. Based on the result with the K356D mutant, we further propose that this anchoring serves to stabilize, compartmentalize and activate SpoIIE, and that these activities are linked through a common feature of SpoIIE.

## The divisome competes with the cell pole for binding of SpoIIE

During sporulation, SpoIIE first accumulates at the polar divisome, constricts along with the septum and then is released into the forespore following the completion of cytokinesis (as shown by time-lapse and structured illumination microscopy in *Figure 1C*, *Videos 1–3*, *Figure 1—figure supplement 1*). This suggests that the divisome competes with the pole for SpoIIE binding and that SpoIIE is not free to associate with the pole until the divisome is disassembled. To investigate this model, we monitored SpoIIE localization in the background of a temperature-sensitive allele of *divIC* that stalls cell division after divisome formation but before cytokinesis (*Levin and Losick, 1994*). In this background, SpoIIE-YFP and ΔTag SpoIIE-YFP localized to the divisome (as visualized with a ZapA-CFP fusion) but not to the cell pole (*Figure 5E*). Similarly, when cytokinesis was blocked during sporulation by a *divIB* deletion (*Thompson et al., 2006*), ΔTag SpoIIE-YFP localized to the divisome but not to the cell pole (*Figure 5E*). Therefore, the divisome sequesters and prevents SpoIIE from associating with the cell pole. We conclude that SpoIIE has affinity for two subcellular sites: the divisome, its dominant binding site, and the pole, where it is captured only after release from the divisome after the completion of cytokinesis.

## SpoIIE is compartmentalized in vegetative cells engineered to divide asymmetrically

The results discussed above suggest a simple model for how SpoIIE and σ^F activity are compartmentalized in the forespore. We propose that SpoIIE is sequestered at the asymmetrically positioned divisome and is released and captured at the proximal (forespore) pole when cytokinesis is completed. In support of this idea, cells that cannot synthesize additional SpoIIE molecules in the forespore nonetheless robustly compartmentalize SpoIIE (*Figure 5—figure supplement 1*). Asymmetric compartmentalization of SpoIIE in the forespore could be achieved by virtue of the

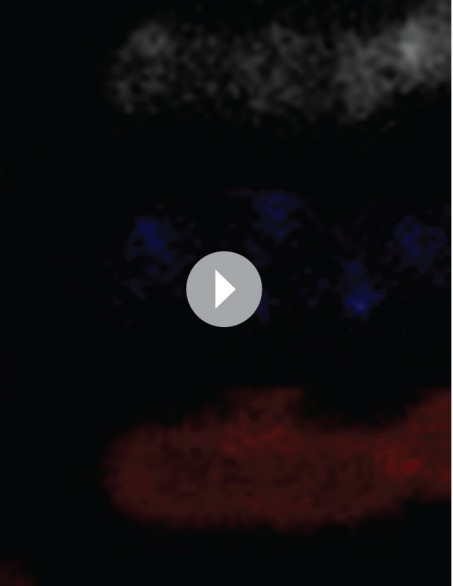

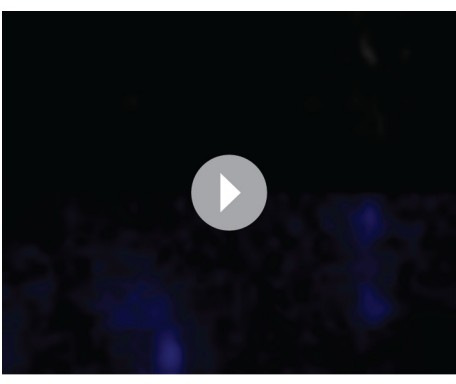

**Video 2.** Movie file of the sporulating cell shown in *Figure 1—figure supplement 1A* (2fps). SpoIIE-YFP is shown in grey, and the divisome marked by CFP-ZapA is shown in blue.

**Video 3.** Movie file of the sporulating cell shown in *Figure 1—figure supplement 1B* (2fps). SpoIIE-YFP is shown in grey, the divisome marked by CFP-ZapA is shown in blue, and the membrane marked by MalFtm-mNeptune is shown in red.

close proximity of the divisome and the forespore pole. Weak SpoIIE association with the pole would be compensated for by the small volume of the forespore and reinforced by protection from degradation by FtsH. Finally, any SpoIIE released into the mother cell would be captured at the divisome, preventing capture at the mother cell pole. Thus, a simple model explains how SpoIIE is protected from degradation and compartmentalized in the forespore only after cytokinesis is complete.

The heart of this model is that asymmetric positioning of the division septum is all that is necessary for compartment specific stabilization and activation of SpoIIE. To test this prediction, we sought to compartmentalize SpoIIE in cells that had been engineered to undergo polar division independently of sporulation. To do so, we artificially expressed *spoIIE* and overexpressed *ftsAZ* in vegetative cells, which was previously demonstrated to reposition the division septum from the mid-cell to near the pole (*Ben-Yehuda and Losick, 2002*). To preclude transcription of sporulation-specific genes, we additionally deleted the master regulator for entry into sporulation, *spo0A*. We then visualized the localization of SpoIIE in these cells. As predicted by our model, cells enriched for SpoIIE were much smaller than average for the entire population (*Figure 6A,B*). Further, to ask whether this compartmentalization of SpoIIE was sufficient to direct cell-specific activation of $\sigma^F$, we additionally induced synthesis of $\sigma^F$ and its regulators SpoIIAA (the anti-anti $\sigma^F$ factor substrate for the SpoIIE phosphatase) and SpoIIAB (the anti-$\sigma^F$ factor) in a strain harboring a reporter for $\sigma^F$ activity. Remarkably, $\sigma^F$ was activated with high selectivity in a subpopulation of the minicells (*Figure 6A,B*). We conclude that polar division is the only feature of sporulation necessary to restrict SpoIIE protein and activity to the small cell and that this is sufficient to explain compartmentalized activation of $\sigma^F$.

## Multimerization of SpoIIE is required for polar anchoring and activation of $\sigma^F$

To gain insight into the molecular mechanism of SpoIIE localization and activation, we expressed and purified the C-terminal cytosolic domain of SpoIIE (residues 320 to 827, SpoIIE$^{320-827}$) for biochemical characterization. A striking feature of SpoIIE$^{320-827}$ was that it multimerizes, and analytical ultracentrifugation revealed these multimers to be hexamers and higher order assemblies of hexamers (*Figure 7A*).

To identify determinants of multimerization and its role for SpoIIE function, we made serial N-terminal truncations of SpoIIE starting at residue 320 and determined the oligomeric state of

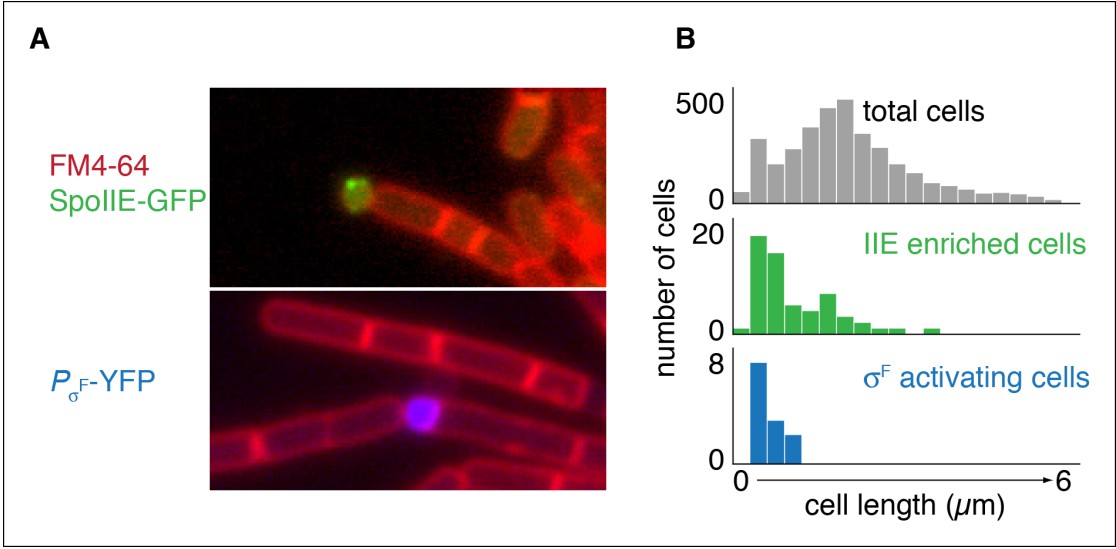

**Figure 6.** Repositioning the septum in vegetative cells is sufficient for compartmentalization of SpoIIE. (**A**) In the top image, vegetatively growing cells producing SpoIIE-GFP and overexpressing the *ftsZ* operon formed minicells enriched for SpoIIE-GFP (strain RL5930). In the lower image, cells additionally expressed the *spoIIA* operon and harbored a YFP reporter for $\sigma^F$ activity (strain RL5931). (**B**) Quantification of images as shown in panel A. SpoIIE enriched cells (representing 56 out of 3168 total cells) are shown in green in the middle plot and are defined as cells with 2 standard deviations above the mean SpoIIE-GFP intensity. Cells that had active $\sigma^F$ were rare in the population (12 cells) and were identified based on detectable levels of YFP fluorescence and their sizes were measured. They are shown in blue at the bottom.

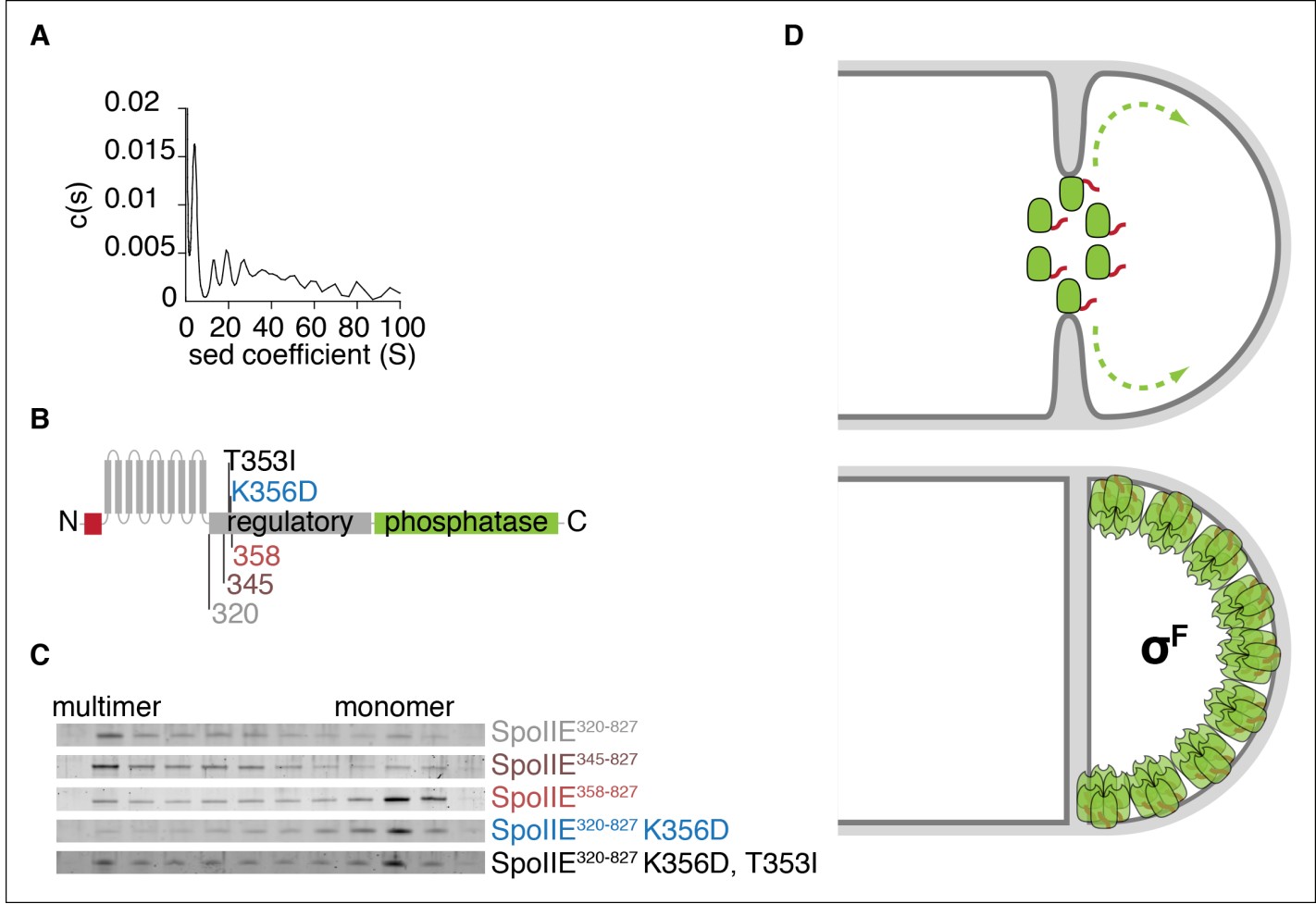

**Figure 7.** Multimerization is required for compartmentalization of SpoIIE and σF activation. (**A**) Purified soluble SpoIIE[320-827] was analyzed by sedimentation velocity analytical ultracentrifugation detected by absorbance at 280nm and fitted to c (**s**) using Sedfit. Peaks corresponding to the predicted sedimentation coefficient for monomeric SpoIIE, hexamers, and multimers of hexamers were observed. (**B**) Diagram of the mutations and truncations in SpoIIE analyzed in panel C. (**C**) Multimerization of SpoIIE variants was analyzed by gel filtration with a 24 ml Superose 6 column. 1 ml fractions from 7-–18 ml of the run were collected, run on SDS-PAGE gels, and stained with SYPRO Ruby. (**D**) Model for handoff of SpoIIE from the divisome to the adjacent cell pole. SpoIIE (green) initially accumulates at the divisome and constricts along with FtsZ during cytokinesis. Prior to the completion of cell division, SpoIIE is degraded by FtsH through its Tag[SpoIIE] (red). (We cannot distinguish whether association with the divisome protects SpoIIE from proteolysis or if SpoIIE turns over while associated with the divisome.) Upon the completion of cytokinesis, SpoIIE transfers to the adjacent cell pole where multimerization protects it from proteolysis (as depicted by the light orange Tags) and leads to phosphatase activation. We propose that close proximity favors transfer to the immediately adjacent pole and that concentration of SpoIIE in the forespore, which is almost entirely derived from the pole, promotes multimerization.

The following figure supplements are available for Figure 7:

**Figure supplement 1.** SpoIIE variants defective for sporulation multimerize.

each by gel filtration (*Figure 7B,C*). We found that the 13-amino acid interval from residues 345 to 358 contained a feature required for multimerization; although SpoIIE truncated to residue 345 (SpoIIE[345-827]) multimerized, all truncations extending to 358 (SpoIIE[358-827]) and beyond did not (*Figure 7C*). This region encompasses K356, raising the possibility that K356D might block stabilization and activation of SpoIIE at the pole by blocking multimerization, and that T35I might suppress the phenotypes of K356D by restoring multimermization. Indeed, gel filtration experiments confirmed that SpoIIE[320-827, K356D] did not multimerize and that T353I partially restored multimerization (*Figure 7C*). Of all the SpoIIE variants analyzed, the K356D substitution uniquely blocked

multimerization, highlighting a key role for the N-terminal region of the regulatory domain in mediating multimerization (*Figure 7—figure supplement 1*).

In sum, these biochemical experiments in conjunction with the in vivo experiments described above lead us to propose that multimerization of SpoIIE is the critical transition that stabilizes SpoIIE, enabling it to recognize the cell pole and leading to its activation as a phosphatase. First, we found that multimerization is required for stabilization of SpoIIE and compartmentalization of SpoIIE to the forespore (*Figure 4B,C—figure supplement 1*). Second, removal of Tag$^{SpoIIE}$ uncoupled multimerization from degradation, and revealed an additional link between SpoIIE activation and multimerization (*Figure 4D*). Finally, we found that recognition of the cell pole by SpoIIE also depended on multimerization, even when SpoIIE was synthesized in vegetative cells uncoupled from degradation and $\sigma^F$ activation (*Figure 5B*).

## Discussion

A hallmark of sporulation is a process of asymmetric division that creates a septum near one randomly selected pole of the cell. Polar placement of the septum directs the protein phosphatase SpoIIE to activate $\sigma^F$ in the resulting forespore. How SpoIIE activates $\sigma^F$ at the right time and in the right place has been one of the enduring mysteries of this developmental system. As the end of the cell used for asymmetric division is chosen without regard to whether it is the old or new pole (*Veening et al., 2008*), the cues that SpoIIE interprets to achieve cell-specific activation of $\sigma^F$ must arise de novo, that is, from the position of the septum rather than from preexisting asymmetry. Indeed, our results show that no feature of the sporulation process other than polar placement of the septum is necessary for compartmentalizing SpoIIE and for cell-specific activation of $\sigma^F$ (*Figure 6*). In addition, as transcription of *spoIIE* commences prior to asymmetric division (*Fujita and Losick, 2003*) (*Figure 2A*), SpoIIE must also respond to temporal cues to ensure it is not active prior to the completion of cytokinesis.

Here we have provided evidence for a model in which SpoIIE leverages the asymmetric position of the septum to selectively associate with the adjacent cell pole of the forespore where it is stabilized and activated (*Figure 7D*). Three key features of our model are as follows:

1. *Capture at the cell pole*. SpoIIE initially accumulates at the polar divisome and is handed off to the forespore pole following cytokinesis. Sequential transfer is enforced by preferential binding to the divisome over the cell pole, and we propose that selectivity for the forespore is achieved by the divisome being immediately adjacent to the forespore pole and that the forespore is largely derived from the pole. Additionally, SpoIIE not captured in the forespore would be sequestered at the distal divisome in the newly formed mother cell, preserving asymmetry.
2. *Spatially restricted proteolysis*. SpoIIE is degraded by the AAA+ protease FtsH and is selectively stabilized in the forespore. This ensures that SpoIIE does not accumulate or become active prior to asymmetric division or in the mother cell following asymmetric division.
3. *Oligomerization*. Polar recognition, protection from FtsH, and activation as a phosphatase are linked by a transition that takes place at the pole and is mediated by oligomerization.

Together these three features provide a simple mechanism for how cues derived from asymmetric cell division restrict SpoIIE to the forespore and couple $\sigma^F$ activation to the completion of cytokinesis. At the same time our model raises several unanswered questions important both for understanding sporulation and diverse related biological systems.

How does SpoIIE localize to the cell pole and the divisome? Localization to the divisome depends on FtsZ and FtsA, the earliest assembling proteins to define the divisome (*Levin et al., 1997*). But whether SpoIIE interacts with these proteins directly, what features of SpoIIE mediate divisome association, and how SpoIIE influences FtsZ polymerization and divisome maturation are unknown. Answering these questions will help us to understand how SpoIIE is transferred from the divisome to the cell pole as well as how SpoIIE influences the position of the division septum. Similarly, localization to the pole depends on DivIVA, which directly senses the shape of the pole and acts as an organizing center for other pole-associated proteins (*Lenarcic et al., 2009*; *Ramamurthi and Losick, 2009*). But it is not known whether this interaction is direct or depends on an accessory protein.

How does oligomerization of SpoIIE promote $\sigma^F$ activation? Genetic and biochemical evidence are consistent with a model in which stabilization, compartmentalization, and activation of SpoIIE are

linked by oligomerization of SpoIIE molecules and that this oligomerization takes place in the forespore after asymmetric division. We cannot exclude the possibility, however, that oligomerization commences earlier in sporulation and that some other unrecognized feature of SpoIIE is additionally required for its transition to a stable and active state in the forespore. Structural information about the organization of SpoIIE oligomers and an in vivo assay for oligomerization may help distinguish between these possibilities and yield new insights into how it contributes to compartment specific $\sigma^F$ activation.

How is activation of $\sigma^F$ coordinated with the completion of asymmetric cell division? Our model proposes two mechanisms to prevent predivisional activation of $\sigma^F$: First, the features of the forespore (small size, high concentration of cell pole, and proximity to the divisiome) that promote SpoIIE stabilization and $\sigma^F$ activation are all emergent properties that depend on completion of cell division. Second, competition between the divisome and cell pole for binding to SpoIIE prevents premature accumulation and activation of $\sigma^F$. Although there has been uncertainty about when SpoIIE is released from the divisome and when asymmetry in SpoIIE compartmentalization is established (*Eswaramoorthy et al., 2014*; *Lucet et al., 2000*; *Wu et al., 1998*), our time-lapse imaging and structured illumination microscopy indicate that SpoIIE constricts along with the FtsZ ring during cytokinesis (*Figure 1*, *Figure 1—figure supplement 1*). This is consistent with our model that SpoIIE remains sequestered at the divisome until cytokinesis is completed. In the future it will be important to determine just how association with the divisome prevents SpoIIE from oligomerizing and activating $\sigma^F$.

How is SpoIIE protected from degradation in the forespore? We have shown that the N-terminal tail of SpoIIE is necessary and sufficient for FtsH-dependent degradation (*Figure 2E,F*) and that stabilization in the forespore is mediated by features of SpoIIE that are required for interaction with the cell pole (*Figure 3,4*). Additionally, we have presented evidence that multimerization of SpoIIE is required for both stabilization and interaction with the pole. One possibility is that multimerization shields the Tag$^{SpoIIE}$ from FtsH as depicted in *Figure 7D*. Alternatively, as FtsH has been shown to have weak unfoldase activity (*Herman et al., 2003*), multimerization might render SpoIIE resistant to FtsH unfolding and hence proteolysis. Finally, although we favor the view that SpoIIE is directly recognized by FtsH, it is conceivable that it requires an adaptor as is the case for some substrates of AAA+ proteases (*Gottesman, 2003*). If so, SpoIIE could be protected from degradation by negative regulation of the adaptor.

How is the phosphatase activity of SpoIIE regulated? Our genetic analysis provides clues for how activation occurs. We found that the V697A substitution locks SpoIIE in a high activity state in vitro, and restores $\sigma^F$ activity in mutants defective for compartmentalization and $\sigma^F$ activation. V697 is in an active site proximal loop (*Levdikov et al., 2011*); in many PP2C phosphatases this loop coordinates a third manganese ion that is critical for activity (*Su et al., 2011*). However, SpoIIE lacks the aspartate that coordinates this manganese, which could indicate that V697A locks the phosphatase in a conformation that compensates for the missing manganese ion. Additionally, we found that the stimulation of phosphatase activity (and binding to the pole) is genetically linked to multimerization: oligomerization, and $\sigma^F$ activation were blocked by the substitution K356D and restored by T353I. We therefore speculate that multimerization induces a conformational change that organizes the catalytic center, compensating for the missing manganese and activating the phosphatase. A test of our hypothesis for a multimerization-dependent conformational change in the active site will require reconstituting multimerization-dependent activation of SpoIIE in vitro. Other PP2C phosphatases, such as the tumor suppressor protein PHLPP (*Gao et al., 2005*), similarly lack the aspartate to coordinate a third magnesium ion, suggesting that our speculation, if correct, could represent a more general regulatory mechanism for PP2C phosphatases.

In summary, we propose that the asymmetrically positioned division machinery – the de novo-generated source of asymmetry – positions SpoIIE to be captured at the adjacent cell pole, triggering $\sigma^F$-directed gene expression in the forespore. Capture at the pole, proteolytic stabilization and stimulation of the phosphatase all depend on oligomerization of SpoIIE. Thus, three interlinked regulatory events are sufficient to explain how SpoIIE exploits a stochastically generated spatial cue to the cell-specific activation of a transcription factor.

## Materials and methods

### Strains and strain construction

*B. subtilis* strains were constructed in PY79 using standard molecular genetic techniques (*Harwood and Cutting, 1990*). Full details of strain genotypes, and construction are provided in *Supplementary file 1*. For IPTG dependent expression ($P_{lac}$), the hyperspank promoter was used (from pDR111a, gift of David Rudner), and for $\sigma^F$ dependent expression ($P_{\sigma F}$), the *spoIIQ* promoter was used. Constructs were made by Gibson Assembly (New England Biolabs, Ipswitch, MA), and point mutations were introduced using QuikChange mutagenesis (Agilent Technologies).

### Isolation of suppressors

Suppressors of the *spoIIE-K356D* mutation were isolated by growing 100 ml cultures of strain RL5895 in DSM sporulation medium at 37°°C for 28 hr. 11 ml of cells were heat killed at 80°C and used to re-inoculate a new 100 ml DSM culture. Heat killed cells from the second round culture were plated on DSM agar plates. Genomic DNA was prepared from the strain and retransformed to strain RL5875, lacking *spoIIE*, to confirm linkage to *spoIIE*, and the *spoIIE* locus was sequenced. Finally, suppressor mutations were then reconstructed by quickchange mutagenesis. Mutations T353I, V697A, V697F, and pseudorevertants K356T and K356Y were each isolated from multiple independent cultures, suggesting that the screen was near saturation.

### Protein degradation and protein levels

Protein degradation rates were measured by shutting off translation by addition of chloramphenicol (100 μg/ml) to cultures. Samples were removed at indicated timepoints and immediately put on ice. Cells were lysed by mechanical disruption in a FastPrep (MP-BIO, Santa Ana, CA). Western blots were conducted by standard procedures and imaged on a BioRad ChemiDoc imager using chemiluminescence. Antibodies used were polyclonal anti-GFP (*Rudner and Losick, 2002*), polyclonal anti-$\sigma^A$ (*Fujita, 2000*), polyclonal anti-DivIVA (*Eswaramoorthy et al., 2014*), and monoclonal anti-FLAG M2 (Sigma Aldrich, St. Louis, MO). Standards were used to determine linearity in each experiment. Immunoprecipitation of DivIVA was performed as published (*Eswaramoorthy et al., 2014*).

### Fluorescence microscopy

All micrographs were acquired on an Olympus BX61 upright fluorescence microscope with a 100X objective, with the exception of timelapse images taken on a Nikon ti inverted microscope. Cells were immobilized on 2.5% agarose pads made with sporulation resuspension medium. To quantitatively analyze micrographs, cells were segmented from phase images using either MicrobeTracker (*Sliusarenko et al., 2011*) or SupperSeggerOpti (*Kuwada et al., 2015*), and analyzed with custom MatLab scripts (scripts for quantitative image analysis are included as a *Source code 1*). Cell specificity of $\sigma^F$ activity was determined by analyzing the distribution of $P_{\sigma F}$-CFP along the long axis of cells. SpoIIE localization profiles were calculated for each cell as the normalized ratio of SpoIIE-YFP to FM4-64 along the long axis of the cell. Because SpoIIE is a transmembrane protein, FM4-64 accounts for differences in membrane area along the cell axis. For sporulating cells that had undergone division, division septa were detected using FM4-64 and cells were oriented based on the position of the polar septum. Vegetative cells (strains RL5930 and RL5931) were induced to produce minicells by dilution from a log phase overnight culture to OD 0.05 and addition of 1mM IPTG, and 0.25% xylose as appropriate. Cells were imaged after 4 hr of growth at 37°C. Segmentation was performed based on phase images subtracted for FM4-64 to identify division septa in chained cells. Cells enriched for SpoIIE-GFP were defined as cells with average SpoIIE-GFP intensity greater than two standard deviations above the mean. Structured illumination microscopy was performed on a Zeiss Elyra microscope in the Harvard Center for Biological Imaging.

### Biochemistry

SpoIIE was expressed as an N-terminal Sumo-6His fusion in BL21(DE3) cells following overnight induction with 0.5 mM IPTG at 14°C. Cells were lysed in 50 mM Tris pH 8.5, 200 mM NaCl, 1mM beta-mercaptoethanol and purified using HisTrap columns (GE Healthcare, Pittsburg, PA) eluting with a gradient of imidazole. The Sumo-6His tag was removed by cleavage with Ulp1 (Sumo

Protease) followed by Ni-NTA subtraction. For velocity analytical ultracentrifugation, SpoIIE was dialyzed to 20 mM Tris pH8.5, 100 mM NaCl, 2 mM DTT overnight and data was collected at 280 nM spinning at 20,000 RPM at The Biophysical Instrumentation Facility (NSF-0070319) at MIT. Data were fit to a continuous model using SedFit (*Schuck, 2000*). Gel filtration was conducted on a 24 ml Superose 6 column (GE Healthcare, Pittsburg, PA), loading 100 µl of 1 µM SpoIIE. Phosphatase assays of soluble fragments of SpoIIE lacking the transmembrane domain (SpoIIE$^{320-827}$ and SpoIIE$^{320-827, V697A}$) were performed using 32P phosphorylated SpoIIAA (phosphorylated by purified SpoIIAB) as a substrate. Multiple turnover reactions were performed with 0.05 µM SpoIIE and varying concentrations of SpoIIAA-P as indicated. Dephosphorylation of SpoIIAA was detected by TLC chromatography on PEI-Cellulose plates developed in 1 M LiCl, 0.8 M Acetic acid.

## Acknowledgements

We dedicate this article to the memory of Patrick J Piggot. We thank T Wilkinson, and I Barak for collaboration and discussions throughout this work, K Ramamurthi and P Eswaramoorthy for discussions and reagents, L Shapiro and J Kardon for valuable comments during manuscript preparation, P Wiggins for code and assistance using SuperSeggerOpti, and A Leech and D Pheasant for assistance with analytical ultracentrifugation.

## Additional information

### Competing interests

RL: Senior Editor, *eLife*. The other author declares that no competing interests exist.

### Funding

| Funder | Grant reference number | Author |
| --- | --- | --- |
| National Institutes of Health | GM18568 | Richard Losick |
| Damon Runyon Cancer Research Foundation | DRG 2051-10 | Niels Bradshaw |

The funders had no role in study design, data collection and interpretation, or the decision to submit the work for publication.

### Author contributions

NB, Conception and design, Acquisition of data, Analysis and interpretation of data, Drafting or revising the article; RL, Conception and design, Analysis and interpretation of data, Drafting or revising the article

## Additional files

### Supplementary files

• Supplementary file 1. Table of strains.

• Source code 1. Matlab scripts.

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
