## [Decision Letter]

Thank you for submitting your work entitled "A Handoff Model for How Asymmetric Cell Division Triggers Cell-Specific Gene Expression in *Bacillus subtilis*" for peer review at *eLife*. Your submission has been evaluated by Michael Marletta (Senior Editor) and three reviewers, one of whom is a member of our Board of Reviewing Editors. The reviewers have discussed the reviews with one another and the Reviewing Editor has drafted this decision.

The reviewers each felt that this was a clearly written manuscript that provided some new insight in the mechanism by which SpoIIE activates σ^F^ in the forespore of developing *B. subtilis*. However, there were several major concerns about the conclusions and model proposed that limited enthusiasm for the paper. After extensive discussion between the reviewers, the consensus was that the paper is not suitable for *eLife* unless the model proposed can be further substantiated through additional experiments and analysis, as summarized below. If each of the issues below can be addressed, we would consider a revised manuscript, subject to an additional round of peer review.

Essential revisions:

1) The results in Figure 3 raised several questions. First, the fact that SpoIIE was apparently active in the mother cell simply by eliminating FtsH-dependent degradation implies that the proposed change in SpoIIE oligomerization during spore formation may not be so critical to its activation. Second, it was noted that the authors haven't shown whether the activation that occurs in mother cells depends on cell division. It has previously been reported that overexpressing SpoIIE bypasses the requirement for cell division (Arigoni et al., 1999), which, with other data, led the authors to suggest that compartmentalization of σ^F^ activity occurred via the exclusion of an inhibitor (perhaps FtsH) from the forespore, so testing dependence is critical. These two issues should be addressed.

2) Central to the model proposed is the notion that SpoIIE's oligomerization state changes following compartmentalization. The authors must show a change in oligomerization state in vivo during sporulation, e.g. by using sucrose gradients to directly monitor oligomerization.

3) It is odd that stabilizing the mutant SpoIIE variants by removing the N-terminal tag does not restore σ^F^ activity. If the only role of oligomerization is to protect SpoIIE from proteolysis, then removal of the degradation tag should bypass the need for oligomerization and restore σ^F^ activity, even if it's no longer compartment-specific. This result suggests that the point mutants have independent defects in activating the phosphatase domain, or that oligomerization itself is necessary for activation. These predictions can be tested in part by analyzing the oligomeric state of their mutants in vitro and by showing oligomerization-dependent protease activity in vitro. In sum, it was not clear whether oligomerization of SpoIIE was protecting it against FtsH-dependent degradation or activating phosphatase activity, or both.

4) The authors need to somehow demonstrate more directly that SpoIIE-YFP at the divisome is transferred to the nearby cell pole. Although the genetics and cell biology presented were consistent with this conclusion, a direct demonstration of the "handoff" being postulated is needed, perhaps by photoactivation and single particle tracking. Additionally, it wasn't clear whether SpoIIE was really being "handed off" or whether SpoIIE potentially just relocalizes from the septum to the pole via release, diffusion, and capture. The word "handoff" implies a more active mechanism, so the mechanism and wording should be clarified throughout the paper.

5) The authors need a more thorough discussion of the SpoIIE literature and an attempt to work towards an integrated model. This issue arose in two contexts. First: SpoIIE requires FtsZ to localize to the septum, but not DivIC. Interestingly, in a *divICts* mutant, there are no septa, but SpoIIE is active, producing dephosphorylated SpoIIAA, but apparently not at a high enough concentration to disrupt the SpoIIAB-σ^F^ complex (Carniol et al. 2004, Feucht 2002). How can these results be reconciled with the model here? Second: Figure 1, Figure 7 and the Introduction (fourth paragraph) state that SpoIIE constricts during cell division. This conflicts with recent data in Eswaramoorthy 2014, which used super resolution microscopy to demonstrate that SpoIIE does not constrict together with the divisome, but rather stays at the edge of the septal disk *and* that SpoIIE is biased towards the forespore side of the septum immediately after cell division and before relocalization to the pole.

Minor points:

1) Does the MalF-TM chimera accumulate properly in the forespore, activate σ^F^, and support wild-type sporulation efficiency? These results would define the contribution of the transmembrane segments to SpoIIE function. If these results have been published previously, they should be mentioned in the Discussion.

2) The authors identified amino acid substitutions in the regulatory domain of SpoIIE which cause sporulation defects and defects in σ^F^ activation, but are not mislocalized. The authors should use these mutants to strengthen the prediction that oligomerization is required for SpoIIE localization in the forespore. In particular, mutants that cannot activate σ^F^ but are properly localized would be expected to oligomerize properly.

3) The SpoIIE variant C399A has very low SpoIIE levels, mislocalized SpoIIE protein, and nearly undetectable transcription driven by σ^F^, yet it only has a ten-fold decrease in sporulation efficiency. The authors should discuss this apparent discrepancy in phenotypes.

4) Figure 1. In the absence of colocalization with FtsZ or with membrane staining, it is unclear how the SpoIIE localization pattern correlates with division during this timelapse.

5) The frequency with which σ^F^ activity is observed in single cells versus in the mother cell of sporangia should be scored and shown.

6) Figure 3. Many of the cells that have uncompartmentalized σ^F^ activity lack sporulation septa. Does activation of σ^F^ in this case depend on cell division? Or does stabilizing SpoIIE lead to σ^F^ activation in predivisional cells? The frequency of the various cell types should be shown.

7) Figure 3. The legend states that "the contrast [for ΔTag-SpoIIE] is approximately 5X brighter for SpoIIE-YFP." The authors must mean that the fluorescence intensity is 5X greater than SpoIIE-YFP.

8) The supplement for Figure 4 should include a few representative sporangia that are stained with FM4-64. The spore titers should be presented in standard scientific notation. 4.2 X108.

9) Figure 6. The graphs showing the frequency with which σ^F^ is activated in minicells does not accurately convey the data. How many total minicells were scored? What fraction had IIE? σ^F^ activity? The data would likely best be presented in table format, so that the precise number of the various types of minicells counted is clear. If it is to be presented in a graphical format, the Y-axis should be labeled and the numbers of each cell type presented.

10) Additional images and quantification is necessary for the cell biological data, so that it is clear how often σ^F^ is activated before polar septation versus active in the mother cell.

11) Introduction, fourth paragraph: what does 'a central regulatory domain' mean? Is it a domain that regulates SpoIIE or that is involved in regulating SpoIIAA? It isn't clear at this point of the paper – eventually there is some evidence that emerges that this domain may affect multimerization status and hence activity of the phosphatase domain, but I think the authors should clarify much earlier.

12) In the subsection “SpoIIE is degraded by FtsH”, second paragraph: Stability of SpoIIE is monitored using a C-terminal FLAG tag. But this tagging could interfere with stability as protein termini often play critical roles in protease activity. This issue should be addressed by examining native SpoIIE stability somehow and by demonstrating the functionality of the tagged SpoIIE.

13) In the subsection “SpoIIE is degraded by FtsH”, end of third paragraph: What's the evidence that degradation is "likely" direct? Additional evidence is needed or the conclusion should end at SpoIIE being degraded in an FtsH-dependent manner. On a related note: the title of Figure 2 ("SpoIIE is degraded by FtsH") and comments in the Discussion (second and fifth paragraphs) should be adjusted given the data shown.

14) Figure 2/E: Should be shown on a semi-log plot.

15) In the subsection “SpoIIE mutants blocked in compartmentalization and σ^F^ activation”, last paragraph: The evidence here that SpoIIE transitions from a hypothetical "active" state to an "inactive" state is weak. It could be that the compartmentalization defective mutants are also disrupted in phosphatase activity, but that phosphatase activity is constitutive. In other words, the mutants identified may prevent SpoIIE from being degraded and from being a proper phosphatase – I don't see the logic that SpoIIE must be transitioning from one state to another in terms of catalytic activity inside the forespore. The simplest test of the authors' idea is to examine in vitro the phosphatase activity of SpoIIE and the various compartmentalization-defective SpoIIE mutants.

16) In the subsection “An allele-specific suppressor of SpoIIE^K356D^”, first paragraph: The 'regulatory domain' still remains ill-defined at this stage of the manuscript and its function/role opaque to all but the *B. subtilis* aficionado.

17) In the subsection “An allele-specific suppressor of SpoIIE^K356D^”, first paragraph: As above (point number 15), it doesn't seem necessary to invoke the notion that SpoIIE's activity as a phosphatase is regulated. It could be that it is just the proteolysis of SpoIIE that is regulated with spore-specific stabilization.

18) In the subsection “An allele-specific suppressor of SpoIIE^K356D^”, last paragraph: Did the authors really test full-length SpoIIE phosphatase activity or a fragment of SpoIIE lacking the TM and potentially other domains? I would have guessed the latter, but the Methods section doesn't provide enough details. If it really is full-length protein, how are the multiple TM helices dealt with?

19) Figure 5: Why does the very bright focus not show up in the line scan? More to the point: this bright focus seems to imply that ΔTag SpoIIE-YFP mainly accumulates away from the pole.

[Editors' note: further revisions were requested prior to acceptance, as described below.]

Thank you for resubmitting your work entitled "Asymmetric Division Triggers Cell-Specific Gene Expression Through Coupled Capture and Stabilization of a Phosphatase" for further consideration at *eLife*. Your revised article has been favorably evaluated by Michael Marletta (Senior Editor), a Reviewing Editor, and one reviewer. The manuscript has been greatly improved but there is still one remaining issue that needs to be addressed before acceptance, as outlined below:

The authors tried, at the reviewers’ request, to generate data supporting the notion that SpoIIE makes a change in oligomeric state during sporulation. As noted in their responses, this did not work and apparently both sucrose gradient fractionation and co-IPs from merodiploid strains indicated that SpoIIE is oligomeric, even in the presence of the K356D mutation. Two points here:

1) Did the authors examine oligomerization via sucrose gradients as a function of development to test whether oligomeric state changes, as postulated? This seems just as important to test as the K356D mutant. And if the experiment was done, it should probably be shown or at least discussed in the paper (see next point as well).

2) The paper, as written, leaves the reader with the feeling that the transition in oligomeric state is known with more certainty than it really is e.g. in the Discussion it states that 'We have presented evidence that multimerization of SpoIIE is a critical transformation'. While I definitely agree that there is some evidence in favor of this model, notably the in vitro studies of oligomerization by a truncated SpoIIE coupled with mutagenesis, there is also potentially evidence against this model or at least insufficient evidence (i.e. the sucrose fractionation experiments noted in the responses to the reviewers) to make a strong conclusion. I think the authors should be more explicit in the Discussion about what the evidence is and they should probably cite the outcome of their sucrose fractionation experiments, both with the K356D mutant and as a function of development. On this topic: I'm also still puzzled why the authors are postulating that the multimerization of SpoIIE occurs in the forespore, implying that cell division and compartmentalization are necessary for this transition, and yet mother cells that can't degrade SpoIIE but haven't yet divided can still show some activation of σ^F^ (Figure 3). Although predivisional cells only constitute 18% of the total cells showing σ^F^ activation in Figure 3, that result still implies, to me, that SpoIIE either multimerizes before septation in some cells or that it doesn't actually undergo a transition. Perhaps I'm being dense here, but this one aspect of the model still doesn't make sense to me and may need a bit of clarification and/or softening of the conclusions and statements about changes in multimerization.

---

## [Author Response]

Essential revisions: 1) The results in Figure 3 raised several questions. First, the fact that SpoIIE was apparently active in the mother cell simply by eliminating FtsH-dependent degradation implies that the proposed change in SpoIIE oligomerization during spore formation may not be so critical to its activation.

Actually, we do conclude that oligomerization is required for both stabilization of SpoIIE and activation of σ^F^. We have modified the text describing Figure 4 (subsection “An allele-specific suppressor of SpoIIE^K356D^”) and Figure 7 (“Discussion”) to make the basis for this conclusion clearer. Specifically, Figure 4 demonstrates that SpoIIE oligomerization is critical for σ^F^ activation even when degradation by FtsH is blocked. Thus, the requirement for oligomerization to stabilize SpoIIE in the forespore can be uncoupled from the requirement for oligomerization to activate σ^F^.

Second, it was noted that the authors haven't shown whether the activation that occurs in mother cells depends on cell division. It has previously been reported that overexpressing SpoIIE bypasses the requirement for cell division (Arigoni et al., 1999), which, with other data, led the authors to suggest that compartmentalization of σ^F^ activity occurred via the exclusion of an inhibitor (perhaps FtsH) from the forespore, so testing dependence is critical. These two issues should be addressed.

We thank the reviewers for drawing our attention to the importance of the connection between cell division and σ^F^ activation and have added data and text that address this link. Specifically, we have added Figure 3 quantifying mis-activation of σ^F^ when FtsH mediated degradation is blocked. The figure shows mis-activation primarily in cells that have completed asymmetric cell division. This supports the conclusion that stabilization of SpoIIE partially uncouples σ^F^ activation from cell division. Partial uncoupling of σ^F^ activation from cell division by stabilizing SpoIIE is consistent with our model that sequestration of SpoIIE at the divisome prohibits premature oligomerization of SpoIIE and activation of σ^F^. Thus, our model is consistent with the results of Arigoni et al., but rather than FtsH being excluded from the forespore (Figure 3—figure supplement 1), we find that competition between binding of SpoIIE to the divisome or the cell pole helps ensure proper timing of σ^F^ activation. We have also added additional text clarifying and expanding this point (subsection “SpoIIE mutants blocked in compartmentalization and σF activation”).

Additionally, we found that blocking cell division with a *divICts* mutation delays but does not block high σ^F^ activity (Figure 8). A simple explanation for these results is that in most cells ∆Tag-SpoIIE is sequestered at the divisome, delaying σ^F^ activation until levels of ∆Tag-SpoIIE saturate the divisome and excess ∆Tag-SpoIIE can oligomerize and activate σ^F^. Predivisional activation of σ^F^ could additionally occur in a subpopulation of cells where *∆tag-spoIIE* is expressed prior to assembly of the divisome (as in Figure 5).

Author response image 1.**DOI:**
http://dx.doi.org/10.7554/eLife.08145.022

2) Central to the model proposed is the notion that SpoIIE's oligomerization state changes following compartmentalization. The authors must show a change in oligomerization state in vivo during sporulation, e.g. by using sucrose gradients to directly monitor oligomerization.

This is the one challenge posed by the reviewers that we failed to address as we now explain. We have performed both sucrose gradient fractionation of SpoIIE and co-immunoprecipitation of SpoIIE from merodiploid strains with differentially tagged SpoIIE constructs. In both cases we detect oligomeric assemblies of SpoIIE even in the presence of the K356D mutation, preventing us from isolating K356D-dependent oligomers. These complexes could be indirect through SpoIIE association with macromolecular complexes such as the divisome, or could be mediated by the transmembrane domain of SpoIIE. Nevertheless, our in vitro studies of SpoIIE oligomerization combined with the unbiased isolation of a suppressor mutation that restored sporulation and oligomerization strongly suggest, we believe, a role for oligomerization in vivo. Furthermore, the additional experiments we performed dissecting the phenotype of the oligomerization mutant and its suppressor provide compelling evidence that oligomerization is required following the completion of asymmetric septation to interact with the cell pole, protect SpoIIE from degradation, and activate σ^F^.

3) It is odd that stabilizing the mutant SpoIIE variants by removing the N-terminal tag does not restore σ^F^ activity. If the only role of oligomerization is to protect SpoIIE from proteolysis, then removal of the degradation tag should bypass the need for oligomerization and restore σ^F^ activity, even if it's no longer compartment-specific. This result suggests that the point mutants have independent defects in activating the phosphatase domain, or that oligomerization itself is necessary for activation. These predictions can be tested in part by analyzing the oligomeric state of their mutants in vitro and by showing oligomerization-dependent protease activity in vitro. In sum, it was not clear whether oligomerization of SpoIIE was protecting it against FtsH-dependent degradation or activating phosphatase activity, or both.

Based on the results in Figure 4 and Figure 5, we do conclude that oligomerization is required both for σ^F^ activation and for stabilization of SpoIIE from degradation by FtsH and have added text to make our reasoning on this point clearer. Responding to reviewer comments here and above in point 1, we now emphasize the connection between oligomerization and σ^F^ activation in our text (subsection “An allele-specific suppressor of SpoIIE^K356D^” and the Discussion). Additionally, we assessed the oligomerization state of other variants of SpoIIE and found that SpoIIE^K356D^ (and truncated variants beginning after amino acid 356) was uniquely defective in oligomerization, consistent with our model that oligomerization is necessary but not sufficient for compartmentalization and activation of σ^F^. These data have now been included as Figure 7—figure supplement 1. We additionally monitored the phosphatase activity of SpoIIE variants in vitro as detailed below in response to point 15.

4) The authors need to somehow demonstrate more directly that SpoIIE-YFP at the divisome is transferred to the nearby cell pole. Although the genetics and cell biology presented were consistent with this conclusion, a direct demonstration of the "handoff" being postulated is needed, perhaps by photoactivation and single particle tracking. Additionally, it wasn't clear whether SpoIIE was really being "handed off" or whether SpoIIE potentially just relocalizes from the septum to the pole via release, diffusion, and capture. The word "handoff" implies a more active mechanism, so the mechanism and wording should be clarified throughout the paper.

In brief, we do and did envision diffusion and capture when we used the word “hand off” and now provide additional data to support this. In light of the confusion it caused, we have added text to clarify this point and we no longer use handoff in the text. Further, and importantly, to demonstrate that SpoIIE is transferred from the divisome to the forespore, we have added Figure 5—figure supplement 1 demonstrating that forespore transcription of *spoIIE* is not required for compartmentalization. To block forespore transcription of *spoIIE,* we prevented forespore capture of the *spoIIE* gene, either by deletion of *racA* (to reduce the efficiency of chromosome capture in the forespore), or by transplanting the *spoIIE gene* to the terminus in cells harboring *spoIIIE36* mutation (to prevent chromosome transport to the forespore).

*5) The authors need a more thorough discussion of the SpoIIE literature and an attempt to work towards an integrated model. This issue arose in two contexts. First: SpoIIE requires FtsZ to localize to the septum, but not DivIC. Interestingly, in a* divICts *mutant, there are no septa, but SpoIIE is active, producing dephosphorylated SpoIIAA, but apparently not at a high enough concentration to disrupt the SpoIIAB-σ^F^ complex (Carniol et al. 2004, Feucht 2002). How can these results be reconciled with the model here?*

To further address the link between cell division and σ^F^ we have added new data as described above and a paragraph to the Discussion (subsection”Fluorescence microscopy”). Specifically, and as in previous studies, we observe SpoIIE accumulation at the divisome but reduced activation of σ^F^ in cells mutant for *divIC*. This is consistent with IIE being largely sequestered at the divisome and not oligomerized at the cell pole. Carniol et al. (2004) saw partial dephosphorylation of SpoIIAA to levels similar to wt sporulating cells but much lower than SpoIIE^V697A^. A simple explanation for the lack of σ^F^ activation in the Carniol et al. (2004) study is that dephosphorylated SpoIIAA is concentrated in the forespore in wild-type cells, while in the *divIC* mutant there is still a significant amount of SpoIIAA-P present throughout the cell.

*Second: Figure 1, Figure 7 and the Introduction (fourth paragraph) state that SpoIIE constricts during cell division. This conflicts with recent data in Eswaramoorthy 2014, which used super resolution microscopy to demonstrate that SpoIIE does not constrict together with the divisome, but rather stays at the edge of the septal disk* and *that SpoIIE is biased towards the forespore side of the septum immediately after cell division and before relocalization to the pole.*

We have added Figure 1 to provide additional evidence of SpoIIE constriction with the divisome and have moved discussion of the constriction of SpoIIE along with the divisome to the Results section. Our results mostly agree nicely with and are complementary to Eswaramoorthy et al. (2014), and we build on their results by providing genetic evidence that association of SpoIIE with DivIVA is required for SpoIIE compartmentalization to the forespore. The major discrepancy as noted is with respect to the co-localization of SpoIIE and FtsZ. The main difference in our experimental approach is that we have taken time lapse images, while Eswaramoorthy et al. looked at static images. Because there is a lag between assembly of the divisome and constriction of the septum, and SpoIIE is released to the forespore while the FtsZ ring is disassembling, it is difficult to accurately stage the static images, which might have obscured the constriction of SpoIIE along with the FtsZ ring. To further document that SpoIIE constricts, we have taken additional time-lapse images of cells with the FtsZ ring and membrane labeled in addition to SpoIIE (Figure 1—figure supplement 1). From these images it is clear that SpoIIE constricts along with the FtsZ ring and is released after the completion of septation. Artifacts from the FM4-64 membrane stain can further complicate efforts to localize SpoIIE at the nascent septum. We imaged SpoIIE during sporulation by structured illumination microscopy, taking advantage of a fluorescent fusion protein to stain the membrane and mark the nascent septum. These images further supported our conclusion that SpoIIE constricts along with the nascent septum.

Minor points:1) Does the MalF-TM chimera accumulate properly in the forespore, activate σ^F^, and support wild-type sporulation efficiency? These results would define the contribution of the transmembrane segments to SpoIIE function. If these results have been published previously, they should be mentioned in the Discussion.

The MalF-TM-SpoIIE chimera does not support wild-type sporulation efficiency and we have added these data to the table in [Supplementary-material SD1-data]. Previous studies (King et al. (1999), Carniol et al. (2004)) that used a MalF-TM-SpoIIE chimera had unknowingly removed the FtsH degradation tag. While stabilized MalF-TM-SpoIIE prematurely activates σ^F^ and prevents asymmetric cell division, MalF-TM-SpoIIE with the FtsH degradation tag is not stabilized in the forespore and fails to activate σ^F^. Thus, the transmembrane domain plays a role in protecting SpoIIE from proteolysis in the forespore.

2) The authors identified amino acid substitutions in the regulatory domain of SpoIIE which cause sporulation defects and defects in σ^F^ activation, but are not mislocalized. The authors should use these mutants to strengthen the prediction that oligomerization is required for SpoIIE localization in the forespore. In particular, mutants that cannot activate σ^F^ but are properly localized would be expected to oligomerize properly.

As discussed above in response to essential revision 3, we have assessed the oligomerization of several additional mutants of SpoIIE and found that SpoIIE^K356D^ was uniquely defective in oligomerization (Figure 1).

3) The SpoIIE variant C399A has very low SpoIIE levels, mislocalized SpoIIE protein, and nearly undetectable transcription driven by σ^F^, yet it only has a ten-fold decrease in sporulation efficiency. The authors should discuss this apparent discrepancy in phenotypes.

The simplest explanation why the *spoIIE^C399A^* mutant has a less severe sporulation defect than other SpoIIE variants is that it supports a low level of σ^F^ activation. This is supported by a longer exposure of the western from Figure 4, which shows a low but significant level of σ^F^-directed expression of a reporter. Additionally, we note that assessment of sporulation efficiency by heat killing is highly sensitive, while assays for IIE levels and σ^F^ activity do not accurately report on events in a minor population of cells. Finally, C399A is part of a motif of four invariant cysteines in SpoIIE, and we observe synergistic effects of mutating multiple cysteines in tandem, suggesting that mutation of a single cysteine does not abolish function.

4) Figure 1. In the absence of colocalization with FtsZ or with membrane staining, it is unclear how the SpoIIE localization pattern correlates with division during this timelapse.

To further document that it does and in response to comment 5 above, we have taken new time lapse images colocalizing SpoIIE and FtsZ, which have been included as Figure 1—figure supplement 1 and as supplementary movies. We observe clear constriction of SpoIIE along with FtsZ and the septal membrane.

5) The frequency with which σ^F^ activity is observed in single cells versus in the mother cell of sporangia should be scored and shown.

These data have been added as Figure 3.

6) Figure 3. Many of the cells that have uncompartmentalized σ^F^ activity lack sporulation septa. Does activation of σ^F^ in this case depend on cell division? Or does stabilizing SpoIIE lead to σ^F^ activation in predivisional cells? The frequency of the various cell types should be shown.

These data have been added as Figure 3.

7) Figure 3. The legend states that "the contrast [for ΔTag SpoIIE] is approximately 5X brighter for SpoIIE-YFP." The authors must mean that the fluorescence intensity is 5X greater than SpoIIE-YFP.

The text of the legend has been clarified to indicate that the contrast has been equalized between the images for display purposes, but that we observe brighter fluorescence signal for ∆Tag-SpoIIE-YFP as expected from the results in Figure 3.

8) The supplement for Figure 4 should include a few representative sporangia that are stained with FM4-64. The spore titers should be presented in standard scientific notation. 4.2 X108.

We have modified the notation of sporulation efficiency in [Supplementary-material SD1-data]. Although we have imaged all of the *spoIIE* mutant strains presented in [Supplementary-material SD1-data], we do not think that images of the defective sporangia would be informative beyond the average traces presented in Figure 4. We have decided not to include this additional data but we would of course agree to do so if the reviewer considers it important.

9) Figure 6. The graphs showing the frequency with which σ^F^ is activated in minicells does not accurately convey the data. How many total minicells were scored? What fraction had IIE? σ^F^ activity? The data would likely best be presented in table format, so that the precise number of the various types of minicells counted is clear. If it is to be presented in a graphical format, the Y-axis should be labeled and the numbers of each cell type presented.

We have added labels indicating the Y-axis scale to the histograms in Figure 6 and have modified the legend to note the total number of cells observed in each case.

10) Additional images and quantification is necessary for the cell biological data, so that it is clear how often σ^F^ is activated before polar septation versus active in the mother cell.

These data have been added as Figure 3.

11) Introduction, fourth paragraph: what does 'a central regulatory domain' mean? Is it a domain that regulates SpoIIE or that is involved in regulating SpoIIAA? It isn't clear at this point of the paper – eventually there is some evidence that emerges that this domain may affect multimerization status and hence activity of the phosphatase domain, but I think the authors should clarify much earlier.

The central regulatory domain is defined physically from sequence analysis (as distinct from the transmembrane domain to the N-terminus and the PP2C phosphatase domain to the C-terminus), and functionally from genetic evidence supporting a role in regulating SpoIIE function. We have modified the text to clearly define our use of the term “regulatory domain” in the Introduction.

12) In the subsection “SpoIIE is degraded by FtsH”, second paragraph: Stability of SpoIIE is monitored using a C-terminal FLAG tag. But this tagging could interfere with stability as protein termini often play critical roles in protease activity. This issue should be addressed by examining native SpoIIE stability somehow and by demonstrating the functionality of the tagged SpoIIE.

We used tagged SpoIIE for degradation experiments due to the fact that C-terminally tagged variants of SpoIIE support sporulation to levels indistinguishable from wild-type (these sporulation efficiency numbers are now included in [Supplementary-material SD1-data]), and YFP tagged SpoIIE is compartmentalized in the forespore. Additionally, native SpoIIE antibodies are of poor quality. In response to the reviewers’ suggestion, we conducted degradation experiments with untagged SpoIIE, and an affinity purified SpoIIE antibody. The results are similar to those seen with the tagged protein, and we detected degradation of SpoIIE that was largely dependent on FtsH. These data have been added to Figure 2—figure supplement 1.

13) In the subsection “SpoIIE is degraded by FtsH”, end of third paragraph: What's the evidence that degradation is "likely" direct? Additional evidence is needed or the conclusion should end at SpoIIE being degraded in an FtsH-dependent manner. On a related note: the title of Figure 2 ("SpoIIE is degraded by FtsH") and comments in the Discussion (second and fifth paragraphs) should be adjusted given the data shown.

We agree that “likely” is too strong and have changed the wording throughout the manuscript, and have modified the title to Figure 2. In the Discussion we present as a model that FtsH degrades SpoIIE, and for simplicity and readability we have left this unchanged.

14) Figure 2/E: Should be shown on a semi-log plot.

Although plotting exponential decay on semi-log axes leads to a linear fit, the linear axes are more intuitive to the general reader, and so we prefer to leave the plots as they are. We have included semi-log plots of the data here for reference.

Author response image 2.**DOI:**
http://dx.doi.org/10.7554/eLife.08145.023

15) In the subsection “SpoIIE mutants blocked in compartmentalization and σF activation”, last paragraph: The evidence here that SpoIIE transitions from a hypothetical "active" state to an "inactive" state is weak. It could be that the compartmentalization defective mutants are also disrupted in phosphatase activity, but that phosphatase activity is constitutive. In other words, the mutants identified may prevent SpoIIE from being degraded and from being a proper phosphatase – I don't see the logic that SpoIIE must be transitioning from one state to another in terms of catalytic activity inside the forespore. The simplest test of the authors' idea is to examine in vitro the phosphatase activity of SpoIIE and the various compartmentalization-defective SpoIIE mutants.

We acknowledge here and in the text that we do not have direct evidence of activation of SpoIIE phosphatase activity. However, this model is consistent with previous evidence (Feucht et al. 2002) and our analysis of the SpoIIE^V697A^ variant. We have measured the in vitro phosphatase activity of a significant set of IIE mutants that do not support σ^F^ activity in vivo and did not detect defects in phosphatase activity for any mutants. This was true for both SpoIIE variants such as K356D that are destabilized and not compartmentalized and for variants such as Q483A that are compartmentalized properly. Importantly mutation of a residue (D628A) that coordinates manganese in the active site abolished phosphatase activity. We suspect that our in vitro phosphatase assay lacks a critical factor for SpoIIE activation (such as DivIVA), and that we are monitoring a low basal level of phosphatase activity. This is further supported by the unphysiologically high manganese concentrations required to detect phosphatase activity for SpoIIE (Km=2mM). It is a major future goal to reconstitute SpoIIE activation in vitro.

*16) In the subsection “An allele-specific suppressor of SpoIIE^K356D^”, first paragraph: The 'regulatory domain' still remains ill-defined at this stage of the manuscript and its function/role opaque to all but the* B. subtilis *aficionado.*

We have modified the text to clearly define our use of the term “regulatory domain” in the Introduction.

17) In the subsection “An allele-specific suppressor of SpoIIE^K356D^”, first paragraph: As above (point number 15), it doesn't seem necessary to invoke the notion that SpoIIE's activity as a phosphatase is regulated. It could be that it is just the proteolysis of SpoIIE that is regulated with spore-specific stabilization.

See response above to point 15.

18) In the subsection “An allele-specific suppressor of SpoIIE^K356D^”, last paragraph: Did the authors really test full-length SpoIIE phosphatase activity or a fragment of SpoIIE lacking the TM and potentially other domains? I would have guessed the latter, but the Methods section doesn't provide enough details. If it really is full-length protein, how are the multiple TM helices dealt with?

As noted in the Methods and the legend to Figure 4—figure supplement 1, phosphatase assays were performed with soluble fragments of SpoIIE encompassing amino acids 320-827. We have added a further note to the Methods section to clarify this.

19) Figure 5: Why does the very bright focus not show up in the line scan? More to the point: this bright focus seems to imply that ΔTag-SpoIIE YFP mainly accumulates away from the pole.

The line scan presented in Figure 5 is an average trace of SpoIIE-YFP fluorescence normalized to FM4-64 from all cells in a population, while the image above is of a representative cell. Some, but not all cells have bright spots of SpoIIE fluorescence that coincide with bright spots of FM4-64 staining. It is not uncommon to observe such artifactual puncta of FM4-64 staining, particularly when cells are stressed, (as they are when cell division is blocked).

[Editors' note: further revisions were requested prior to acceptance, as described below.]

The authors tried, at the reviewers’ request, to generate data supporting the notion that SpoIIE makes a change in oligomeric state during sporulation. As noted in their responses, this did not work and apparently both sucrose gradient fractionation and co-IPs from merodiploid strains indicated that SpoIIE is oligomeric, even in the presence of the K356D mutation. Two points here:1) Did the authors examine oligomerization via sucrose gradients as a function of development to test whether oligomeric state changes, as postulated? This seems just as important to test as the K356D mutant. And if the experiment was done, it should probably be shown or at least discussed in the paper (see next point as well).

Indeed, we did examine oligomerization as a function of developmental stage using sucrose gradients (comparing time points during sporulation and genetic perturbations to arrest development at various stages). Due to the technical limitations mentioned above, these experiments were uninformative either way. As described below, we have added a paragraph that to the Discussion that addresses the uncertainty in when oligomerization commences.

2) The paper, as written, leaves the reader with the feeling that the transition in oligomeric state is known with more certainty than it really is e.g. in the Discussion it states that 'We have presented evidence that multimerization of SpoIIE is a critical transformation'. While I definitely agree that there is some evidence in favor of this model, notably the in vitro studies of oligomerization by a truncated SpoIIE coupled with mutagenesis, there is also potentially evidence against this model or at least insufficient evidence (i.e. the sucrose fractionation experiments noted in the responses to the reviewers) to make a strong conclusion. I think the authors should be more explicit in the Discussion about what the evidence is and they should probably cite the outcome of their sucrose fractionation experiments, both with the K356D mutant and as a function of development.

In our revised manuscript we have added a paragraph to the Discussion (fourth paragraph) that directly discusses these issues and have changed the language in the seventh paragraph. In particular, we discuss an alternative model in which oligomerization commences earlier in sporulation and state that we cannot distinguish the models without an in vivo assay for oligomerization.

On this topic: I'm also still puzzled why the authors are postulating that the multimerization of SpoIIE occurs in the forespore, implying that cell division and compartmentalization are necessary for this transition, and yet mother cells that can't degrade SpoIIE but haven't yet divided can still show some activation of σ^F^ (Figure 3). Although predivisional cells only constitute 18% of the total cells showing σ^F^ activation in Figure 3, that result still implies, to me, that SpoIIE either multimerizes before septation in some cells or that it doesn't actually undergo a transition. Perhaps I'm being dense here, but this one aspect of the model still doesn't make sense to me and may need a bit of clarification and/or softening of the conclusions and statements about changes in multimerization.

I think the reviewers are overlooking the fact that SpoIIE in these cells accumulates to high levels because we have removed the degradation tag. Figure 5 shows that when stabilized, SpoIIE accumulates at the poles of predivisional cells, which should lead to predivisional activation of σ^F^. Although we cannot make a direct comparison (because σ^F^ activation perturbs SpoIIE localization), the frequency with which we observe polar SpoIIE is consistent with the fraction of predivisional cells with σ^F^ activity.